# CAUSAL INTERPRETATION OF NEURAL NETWORK COMPUTATIONS WITH CONTRIBUTION DECOMPOSITION

**Joshua Brendan Melander**[‡]
Department of Neurobiology
Stanford University

**Zaki Alaoui**[‡]
Department of Neurobiology
Stanford University

**Shenghua Liu**[‡]
Department of Physics
Stanford University

**Surya Ganguli**
Department of Applied Physics
Stanford University

**Stephen A. Baccus**
Department of Neurobiology
Stanford University

## ABSTRACT

Understanding how neural networks transform inputs into outputs is crucial for interpreting and manipulating their behavior. Most existing approaches analyze internal representations by identifying hidden-layer activation patterns correlated with human-interpretable concepts. Here we take a direct approach to examine how hidden neurons act to drive network outputs. We introduce CODEC (**Co**ntribution **Dec**omposition), a method that uses sparse autoencoders to decompose network behavior into sparse motifs of hidden-neuron contributions, revealing causal processes that cannot be determined by analyzing activations alone. Applying CODEC to benchmark image-classification networks, we find that contributions grow in sparsity and dimensionality across layers and, unexpectedly, that they progressively decorrelate positive and negative effects on network outputs. We further show that decomposing contributions into sparse modes enables greater control and interpretation of intermediate layers, supporting both causal manipulations of network output and human-interpretable visualizations of distinct image components that combine to drive that output. Finally, by analyzing state-of-the-art models of neural activity in the vertebrate retina, we demonstrate that CODEC uncovers combinatorial actions of model interneurons and identifies the sources of dynamic receptive fields. Overall, CODEC provides a rich and interpretable framework for understanding how nonlinear computations evolve across hierarchical layers, establishing contribution modes as an informative unit of analysis for mechanistic insights into artificial neural networks.

## 1 A FRAMEWORK FOR UNDERSTANDING BIOLOGICAL AND ARTIFICIAL NEURAL NETWORKS

Biological and artificial neural networks both produce computations using cascading nonlinear operations that do not lend themselves to simple interpretations. Despite the widespread study and use of neural networks, there is no standardized framework to understand how a given network output is generated from its input through its intermediate stages. Understanding the mechanisms by which networks behave promises to accelerate studies of the nervous system, lead to more effective design of efficient networks, reveal general principles of information processing in complex systems, and is also important for guiding the development of safe AI systems (Murdoch et al. (2019); Doshi-Velez and Kim (2017); Lipton (2017); Rudin (2019)).

An essential aspect of both artificial and biological neural networks is that their behavior is created by sets of internal components. The question we approach here is: *How do the components of a network act together to construct the output from the input?*

---

[‡]These authors contributed equally to this work.

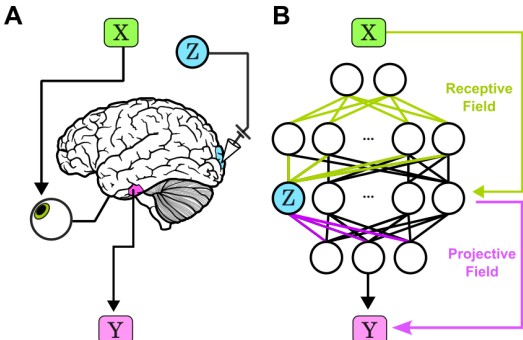

Figure 1: **Understanding the contribution of an intermediate component to downstream computation.** (A) Biological and (B) artificial neural circuits construct computations by combining sets of upstream components in an input-dependent manner. The action of a network component $Z$ is a composition of its receptive field, or sensitivity to input $X$, and its projective field, or effect on output $Y$. Measuring both is required to explain how the intermediate component contributes to the overall behavior of the system.

Sensory systems provide a natural framework for this question, with a well-defined input-output structure, clearly described tasks, and hierarchical architectures that inspired the development of convolutional neural networks (CNNs) (Fukushima (1980)).

A neural network unit transforms inputs through its *receptive field*, and influences downstream processing through its *projective field* (Lehky and Sejnowski (1988)). The composition of these transformations defines a unit's *contribution* to the system's output, which is a direct causal measure (Tanaka et al. (2019); Maheswaranathan et al. (2023)).

Most interpretability methods focus on unit activations, which reflect only the receptive field stage. Causal influence of a unit must then be demonstrated via ablations or perturbations. Direct analyses of contributions, rather than activations, offer a more principled route, and contribution analysis in CNN models of the retina has led to specific hypotheses for how neural cell types perform specific functions (Tanaka et al. (2019); Maheswaranathan et al. (2023)). Yet existing approaches studying contributions have not examined how groups of units act together. This is a significant limitation, as both biological (Olshausen and Field (2004)) and artificial networks compute through coordinated population activity. In CNNs, the output of feature-selective populations can be combined to classify images (Rawat and Wang (2017)). In the visual system, the parallel parvocellular and magnocellular pathways roughly support chromatic and high acuity vision vs. achromatic vision and motion sensitivity (Solomon (2021)), and recent neuroscientific theories suggest that groups of retinal ganglion cells operate in structured modes with specific computational roles (Prentice et al. (2016)). Understanding a network thus requires not just identifying features in the internal representation, but also explaining how combinations of those features are used to construct different outputs.

### EXISTING TOOLS FOR INTERPRETING ANNS

For artificial neural networks (ANNs) with millions of parameters, identifying causal structure is challenging because computation arises from nonlinear interactions, making the effects of single units strongly dependent on the input and activity of other units. In both biological and artificial networks, much attention has been devoted to studying latent representations at intermediate stages in the network by analyzing network activity. Although these methods have found patterns in activations via clustering or sparse autoencoders (Fel et al. (2023)), analyzing representations fundamentally does not address the causal question of how internal elements act to influence the output.

More recently, methods like Integrated Gradients, SmoothGrad, and Grad-CAM have focused on revealing the influence of inputs on model outputs via saliency maps (Selvaraju et al. (2020); Smilkov et al. (2017); Sundararajan et al. (2017)). However, these approaches offer limited insight into the actions of intermediate stages. Visual components such as edges and textures may be needed to dis-

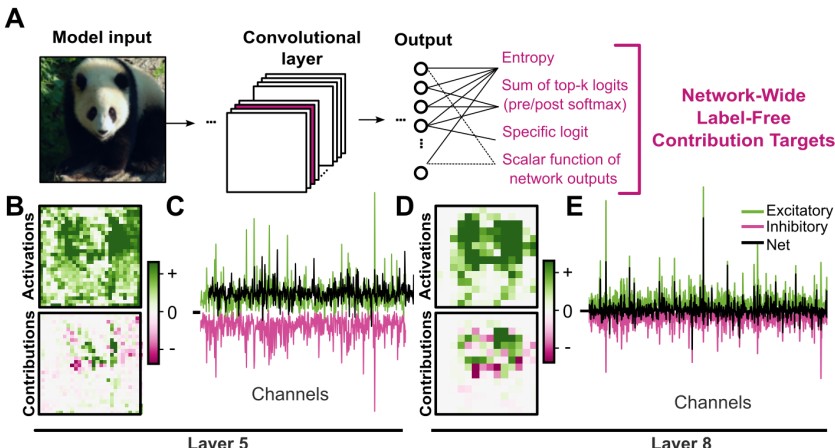

Figure 2: **Hidden-neuron contributions in a deep convolutional network.** (A) Pipeline of computing contributions for an image processed through ResNet-50. Gradient-based attribution methods are extended to compute the contributions of each hidden unit to scalar targets of network output such as entropy of logits, sum of top-$k$ logits, and individual class logits. (B) Spatial map of activations and contributions of a single channel in Layer 5. (C) Mean positive, negative and net contribution for each channel. (D–E) Same as (B–C) for Layer 8.

criminate objects, but those same features in other locations may be unrelated to the target object yet still represented in hidden activations. Thus, compared to the analysis of activations, contribution analysis distinguishes between building blocks of computation that causally drive the output and those that are irrelevant. Accordingly, a key open problem is to characterize how networks causally integrate distributed latent features across channels and neurons to generate outputs, analogous to how biological networks produce functional effects through circuit interactions.

A NEUROSCIENCE-INSPIRED ANN INTERPRETABILITY FRAMEWORK

We introduce a method for analyzing how intermediate neurons drive network output. The first step extends attribution techniques such as Integrated Gradients to measure contributions in internal layers. We compute each hidden neuron's contributions across all stimuli, capturing the combined effects of their receptive and projective fields. These contributions represent the actions neurons take to construct the output.

Next, we decompose these contributions across inputs into a set of modes reflecting coordinated neuronal actions, an approach we call **contribution decomposition (CODEC)**. Unlike analyses of activations, CODEC directly captures causal effects on outputs and can be applied to any trained feedforward model without access to training data or labels. This general-purpose framework quantifies how groups of neurons drive behaviors. In the context of image recognition, the receptive fields of hidden neurons define visual features that are building blocks, and the contribution modes are the assembly instructions that show how those components are used to construct classes.

CODEC is a general framework composed of different stages that can be adapted for artificial and biological neural networks:

1. **Contribution target:** The specific output neuron or behavior (a scalar function of output neurons) whose computational basis we wish to understand.

2. **Contribution algorithm:** Quantification of how each hidden unit contributes to the target output for a given input.

3. **Decomposition of contributions:** The core computational modes representing common patterns of how neurons act together across inputs and outputs.

4. **Visualization in input space:** Contribution mapping to reveal how the input features that drove the key channels or neurons within each mode are used for output identification.

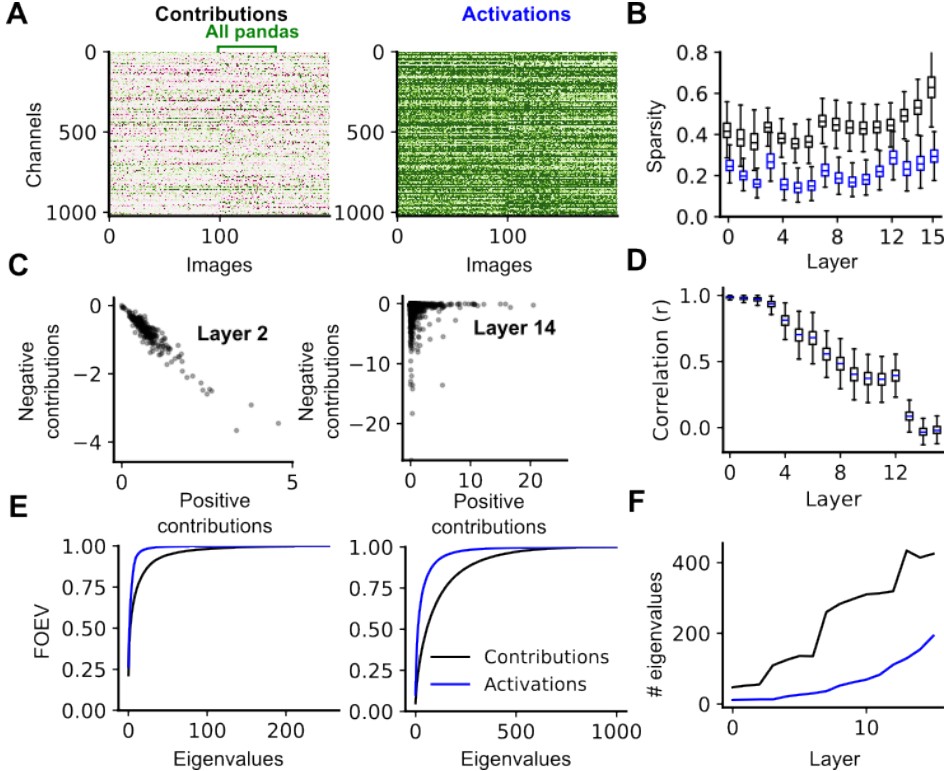

Figure 3: **Channel contributions through the network become more sparse, single-signed and high dimensional.** (A) Example matrix of spatially-summed contributions and activations from one layer for all channels and images from four classes. (B) Hoyer sparsity index for contributions and activations across network layers. (C) Scatter plot of mean negative vs. mean positive contributions of each channel to network output. (D) Correlation coefficient between positive and negative contributions of individual channels across network depth. (E) Fraction of explained variance (FOEV) across all class-averaged channel weightings for layers 2 and 14. (F) Number of components required to reach 95 percent FOEV.

These methods establish a new interpretability tool that examines the intermediate neurons in a network, identifies what input features they are sensitive to and their individual effects on network output, and reveals how their combined actions ultimately influence the network's behavior.

## 2 MEASURING CONTRIBUTIONS OF HIDDEN-LAYER NEURONS

The contribution of a hidden neuron to network output is a composition of its overall input and its overall output (Fig. 1) , and several methods have been used to calculate such effects. Integrated Gradients has most commonly been applied from network output to input (Sundararajan et al. (2017)), but have also been applied to analyze the effects of hidden neurons in models of biological networks (Tanaka et al. (2019); Maheswaranathan et al. (2023)). An alternative attribution method, ActGrad (Selvaraju et al. (2020)), is defined as the element-wise product of activations and gradients (ActGrad$_j$ = $h_j \cdot \frac{\partial y}{\partial h_j}$) where $h_j$ is the activation of hidden unit $j$, but is prone to noisy fluctuations when evaluated locally.

Formulating a single contribution target was crucial to avoid intractable 3D decompositions, which would yield an array of shape $n_{\text{inputs}} \times n_{\text{neurons}} \times n_{\text{logits}}$ for each layer if contributions were computed separately for each logit. We found that computing contributions to the top logit for a given input was sufficient for an accurate reconstruction. Additional scalar objectives are offered in the code: (1) the sum of top-$k$ output logits, reflecting the model's confidence in its strongest predictions, and (2) the entropy of the output distribution, measuring prediction uncertainty.

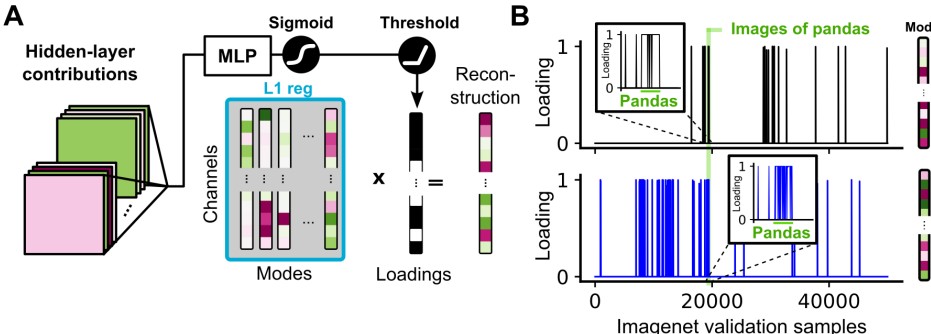

Figure 4: **Sparse autoencoder decomposition of network contributions.** (A) Schematic of contribution decomposition. Channel contributions are spatially summed and an autoencoder is trained to reconstruct the matrix of contributions by images by creating modes with sparse loadings (the weighting for each mode for particular image). Loadings are passed through a sigmoid then a threshold, and regularized to encourage sparsity. (B) Loadings from the mode that maximally correlated with the class "panda" for contributions (top, black) and activations (bottom, blue). Inset shows the loadings for 50 images of panda and 100 images of other classes.

CONTRIBUTIONS OF CONVOLUTIONAL LAYERS IN IMAGE CLASSIFICATION NETWORKS

Biological visual systems, such as the retina, primary visual cortex, and inferotemporal cortex, share many architectural features with CNNs (Yamins et al. (2014)). Thus, we characterized the computational structure of each layer in benchmark CNNs such as ResNet-50 (He et al. (2016)) by analyzing neuronal contributions.

Starting with an input image, we compute the contributions of all hidden neurons to the scalar target (Fig. 2A). We varied contribution algorithms (ActGrad, Integrated Gradients, SmoothGrad) and targets (e.g., top logit, top-$k$ logit sum, entropy), and found that contributions were consistently spatially sparse compared to activations (Fig. 2B,D, Supplementary Fig. S1A). A key property of Integrated Gradients is completeness: contributions sum to the scalar output target. Thus, spatially summing contributions within a channel gives its net effect on the prediction, enabling assignment of a single contribution value per channel, or cell type (Fig. 2C). Applying this procedure across the entire 50,000 ImageNet (Deng et al. (2009)) validation images produced matrices of channel contributions for selected network blocks throughout ResNet-50, with each matrix having dimensionality $d$ channels × 50,000 images (Fig. 3A). For the remaining analyses, we used Integrated Gradients to the top logit with 10 integration steps as our standard contribution method.

## 3 LAYERWISE EVOLUTION OF NEURAL CONTRIBUTIONS IN CNNS

To examine how the actions of the network evolve throughout its layers, we computed for the contributions of each channel the Hoyer sparsity index, a normalized measure that computes the ratio of L1 to L2 norms, and ranges from 0 (all channels equally active) to 1 (only one channel active). At all layers, contributions consistently showed high sparsity across channels than activations, indicating that only a small subset of channels are functionally relevant for each classification decision (Fig. 3B). Furthermore, contributions increased in sparsity throughout the network, aligning with the intuition that feature selectivity emerges with hierarchical depth.

Hidden unit activations are constrained by ReLU nonlinearities to be positive. However, contributions can be positive or negative, indicating whether a spatial position increases or decreases the likelihood of the target output. Thus, a highly active unit in the activation map can be used to inhibit the network's output, as revealed by its contribution. These opposing influences, similar to excitatory and inhibitory interactions in biological vision (e.g., on-off receptive fields), are essential to neural computation but hidden in activations. We therefore examined the relationship between positive and negative contributions across layers in ResNet-50. Importantly, contribution sign reflects the net impact on network output, not the polarity of synaptic weights.

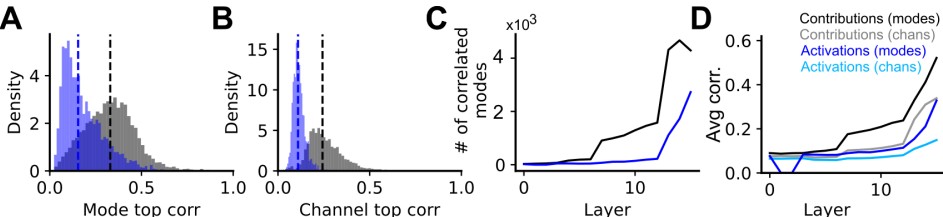

Figure 5: **Emergence of meaningful contribution modes in intermediate layers.** (A) Histograms of each mode's maximum correlation with binary class indicators for contributions (grey) and activations (blue) at hidden layer 13. (B) Same as (A) for the correlation of individual channel contributions or activations with class indicators. (C) Number of modes with a correlation to a class of greater than 0.2. (D) Mean of the maximal class-correlation as a function of layer.

To compare positive and negative contributions, we computed the contribution at each location (Fig. 2B–E), and then separated the contribution into positive and negative components prior to spatial summation. We found that in channels of earlier layers, the magnitude of positive and negative contributions were highly correlated within each channel. However, through the network, positive and negative contributions became progressively decorrelated (Fig. 3C–D). One possible explanation for this shift is that lower-level features such as edges and textures encoded in earlier layers exhibit strong spatial correlations, necessitating that individual channels contribute both positively and negatively to remove these correlations, similar to retinl computations such as center-surround receptive fields (Pitkow and Meister (2012)) and object motion sensitivity (Ölveczky et al. (2003)).

A common technique for identifying class-correlated latent dimensions is to average activations over many examples of single class (Kim et al. (2018)).We thus examined the dimensionality of spatially summed and class-averaged activations and contributions using principal component analysis. Both quantities increased in dimensionality through the network, but contributions showed higher dimensionality than activations as measured by eigenvalues needed to reach 95% variance (Fig. 3E–F). Throughout the network, contributions increased in sparsity and dimensionality, and decorrelated their positive and negative effects.

## 4 DECOMPOSING CONTRIBUTIONS INTO COMPUTATIONAL MODES

To uncover structure within these contributions, we decomposed them into a set of modes using an autoencoder consisting of an encoder network $f_{\text{enc}} : \mathbb{R}^d \to \mathbb{R}^k$, and a non-negative dictionary $\mathbf{D} \in \mathbb{R}_+^{d \times k}$, where $k$ is the number of modes. Typically $k = N \cdot d$ for an overcomplete representation with expansion factor $N$. Each column $\mathbf{m}_i \in \mathbb{R}^d$ of $\mathbf{D}$ defines one mode. Given contributions $\mathbf{c} \in \mathbb{R}^d$, the encoder computes pre-activation loadings $\mathbf{z}_{\text{pre}} = f_{\text{enc}}(\mathbf{c})$. Sparsity is enforced by hard thresholding: $z_i = z_{\text{pre},i}$ if $z_{\text{pre},i} \geq \tau$, and zero otherwise. The reconstruction is $\hat{\mathbf{c}} = \mathbf{D}\mathbf{z}$, and the autoencoder is trained to minimize the loss $\mathcal{L} = \|\mathbf{c} - \hat{\mathbf{c}}\|_2^2 = \|\mathbf{c} - \mathbf{D}\mathbf{z}\|_2^2$, with optional L1 regularization applied to the loadings and modes. Non-negativity constraints are imposed on the dictionary $\mathbf{D}$. Decomposition of contributions resulted in a set of $k$ modes of dimension $d$, and a set of $n$ loadings for each mode reflecting the weighting of those modes for each image that reconstructed the matrix of contributions with high accuracy, (average $R^2 = 0.85$ for contributions and $0.84$ for activations across all layers, Supplementary Fig. S2A). Training takes $\sim$3–7 minutes per layer for 200 epochs on an NVIDIA RTX 6000 GPU (see Supplementary Sec. 10.4 for details). Our baseline model architecture used a threshold of 0.9 and a dictionary size 3 times the number of channels being learned. Models were trained for 300 epochs with a learning-rate of 5e-5 and batch-size of 128. L1 regularization was added to the dictionary with a scaling factor of 5e-5.

To measure how closely related these modes were related to specific network outputs, we correlated the loadings over the entire dataset (50,000 validation images) with a binary vector indicating whether a given image belonged to a given class (Fig. 4B). This resulted in a $k$ by 1000 correlation matrix, representing the correlation of each mode with each ImageNet category. We found that contribution modes were more correlated with Imagenet classes than modes recovered using activations, in particular at intermediate layers (Fig. 5A–C, Fig.S9). In addition, we found that

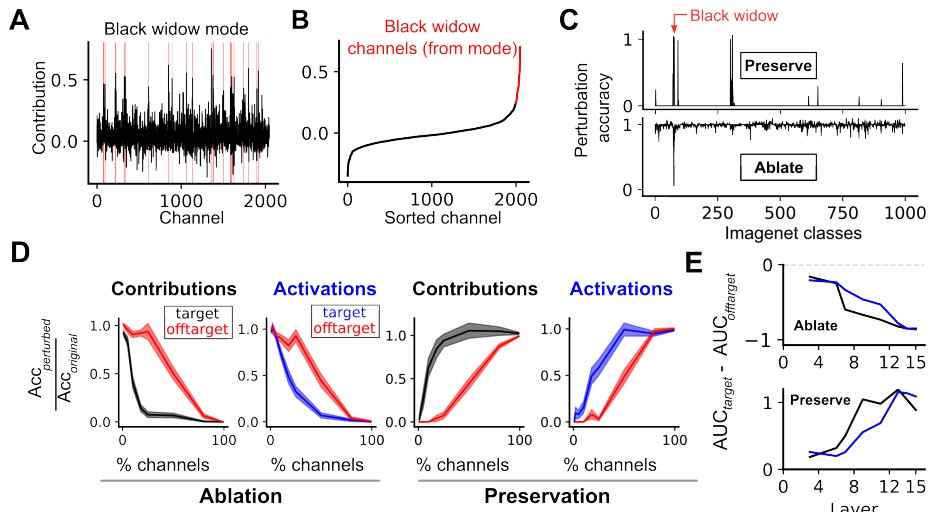

Figure 6: **Contribution-guided network control.** (A) Example mode most correlated with the "black widow" class, showing channel weightings. (B) Sorted channel weightings from (A) with a few top channels highlighted for selection. (C) Results for "black widow" classification: Top shows preservation analysis (accuracy when keeping only selecte channels), bottom shows ablation analysis (accuracy when removing selecte channels). (D) Normalized accuracy change for the target class when ablating (left) or preserving (right) channels from the most correlated contribution mode (black), activation mode (blue), or random class (red, representing non-target/off-target performance). (E) Performance score quantifying perturbation effectiveness as the area between target and off-target curves from (D) across all blocks, normalized to the off-target area.

contribution modes, despite not having access to class labels during optimization, were more correlated with classes than were individual channels. This indicates the success of CODEC at revealing patterns of combined channel contributions that had relevance to specific network outputs (Fig. 5D).

Results were robust to a range of variation to SAE hyperparameters. SAE reconstruction performance was affected little by L1 regularization, random seed, and whether modes were constrained to be positive. SAE $R^2$ was affected only when threshold was too high (0.9 vs 0.5), dictionary size was equal to or smaller than the number of channels, or MLP size was not considerably greater than the number of classes (Supplementary Figs. S3 and S4). These results indicate that SAE performance is robust provided these hyperparameters are kept sufficiently away from their boundary regimes.

## 5    CONTROLLING NETWORK BEHAVIOR USING CONTRIBUTION MODES

Contribution modes represent the coordinated causal effects on network output. To examine how these effects could be used to control ImageNet classification, we perturbed ResNet-50 by targeting channels identified through CODEC analysis. We identified the mode most correlated with each class, then measured classification accuracy under two conditions: **ablation** (removing the top-weighted channels) and **preservation** (retaining only those channels). We quantified these effects across all 1000 ImageNet classes by calculating the ratio in accuracy for each class between perturbed and unperturbed networks. For the "black widow" class, ablating 2% of salient channels identified from the top 2 most correlated modes greatly reduced target-class accuracy while leaving off-target classification performance largely unaffected.(Fig. 6C (bottom)). Additionally, preservation analysis yielded networks that could accurately classify only the targeted class (Fig 6C (top)). We randomly sampled the ImageNet validation dataset and compared perturbation performance for a given mode and target class, and a random off-target class, while varying the percentage of channels perturbed. Targeting channels by contribution modes more reliably identified necessary and sufficient channels for classification compared to activation-based analyses, requiring fewer channels to completely ablate target-class computation (Fig. 6D). A sharp increase in ablation efficacy was

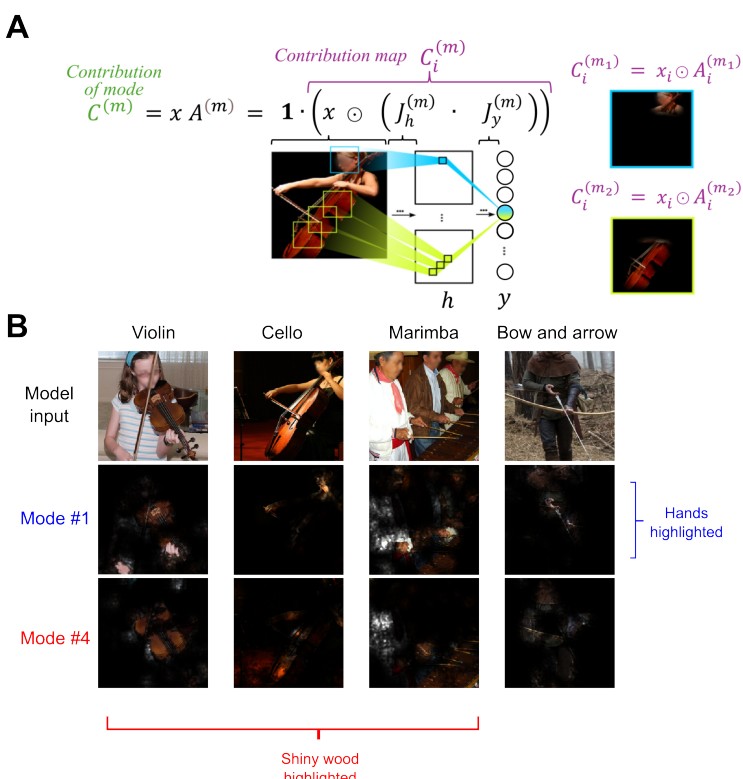

Figure 7: **Visualizing stimuli acting through contribution modes.** (A) Contribution mapping reveals how parallel pathways extract different information to drive network output. The contribution map is an intermediate stage for computing contributions prior to input summation defined by observing that the inner product of the image $x$ and sensitivity $A$ can be expanded into a product of **1**, the vector of ones, and an elementwise product that is the contribution map. Symbols are shown as their transpose to match the left to right schematic. (B) Visualization of 4 example stimuli from different classes through two modes highly correlated with violin at layer 7 of Resnet-50. The information conveyed by modes are interpretable features that can be identified across class and stimuli (shiny wood, hands, etc.). To preserve original colors, maps are shown averaged over color channels and as a weighted mask on the original image, though they contain color information not shown.

observed between blocks 6 and 7, potentially suggesting a shift in how semantic information is represented at this depth (Fig. 6E). Additionally, we demonstrated that our approach generalizes beyond the 1000 ImageNet classes, successfully ablating / preserving specific taxonomic categories, indicating that CODEC can identify non-labled computational pathways even within broader semantic groupings (Supplementary Fig. S10).

# 6 VISUALIZING INPUTS THAT CAUSE HIDDEN CONTRIBUTIONS

Attribution methods such as Integrated Gradients compute gradients from outputs to inputs to identify regions most influential to a decision. We extend this idea to hidden channels: what features of the input does a channel or mode use to drive the output for a given image? Using CODEC-selected salient channels, we isolated the gradient pathway from outputs to inputs that passes only through channels of interest, decomposing traditional input-output saliency maps into interpretable, channel- and mode-specific contributions. For each channel $c$ of mode $m$ and each spatial location $p$, the input sensitivity is $A_i^{(c,p)} = J_{y,h_{c,p}} J_{h_{c,p},x_i}$, where $h_{c,p}$ is the activation at position $p$ in channel $c$, $J_{y,h_{c,p}} = \frac{\partial y}{\partial h_{c,p}}$ and $J_{h_{c,p},x_i} = \frac{\partial h_{c,p}}{\partial x_i}$. Aggregating over all selected channels in a mode and positions yields $A_i^{(m)} = \sum_{c \in m} \sum_p A_i^{(c,p)}$. Whereas $A_i^{(m)}$ captures a spatial map of output sensi-

tivity to pixel $i$, the *contribution* map viewed in input space reflects the cumulative effect of input pixels on the output, approximated as the sensitivity weighted by the input: $C_i^{(m)} = A_i^{(m)} \odot x_i$. The spatial sum $\sum_i C_i^{(m)}$ recovers the contribution of the mode, $C^{(m)}$ . Contribution mapping highlights regions that most strongly drive contributions within the mode's most relevant channels. Fig. 7 shows example visualizations. Additional methods are described in Supp. Sec. 10.7, among which the method described here is InputGrad. The contribution map shows pixels used by hidden channels within modes to drive network output for different classes. Masking the original image with contribution maps shows that a given mode computes the same lower-level semantic features across images from different classes (Fig. 7). We expect that further analysis of the structure of these modes in input space will reveal meaningful visual components of objects used to construct visual classes.

## 7 INTERPRETING BIOLOGICAL NEURAL NETWORK MODELS WITH CODEC

We then used contribution decomposition to interpret the structure of computation in the early visual system. Previous results (Maheswaranathan et al. (2023)) have shown that three layer CNN models capture the responses of retinal ganglion cells to natural scenes, and are interpretable in that hidden units are highly correlated with recordings from retinal interneurons not used to fit the model. We therefore applied CODEC to these CNN models to interpret how groups of model interneurons contributed to retinal output.

The CNN model consisted of 3 layers, with 8 channels (cell types) in the first two layers, yielding a response of 4–17 recorded ganglion cells (Fig. 8A). In order to choose a single contribution target, rather than choosing the top-1 logit as we did in the case of the image classification network, we chose the surprisal, or self-information, $I(x) = -\log_2 P(x)$, an information theory measure that reflects how unexpected a response is. Estimated from the covariance of the output, $I(x)$ varies with the Mahalanobis distance of the population response from the mean, $(\mathbf{x} - \boldsymbol{\mu})^\top \boldsymbol{\Sigma}^{-1} (\mathbf{x} - \boldsymbol{\mu})$, plus a constant term, where $\boldsymbol{\Sigma}$ is the covariance matrix and $\boldsymbol{\mu}$ is the mean response. With this target, positive contributions are those that create a more unexpected response.

CODEC identified modes in the first two layers that combined to drive cells at different times, as measured by computing the correlation between cell firing rates and mode loadings (Fig. 8B-C). We then clustered cell types by the average pattern of active modes that drove the cell. Clustering the mode pattern in layers 1 and 2 yielded very similar results, indicating the robustness of identifying pathways that drove different cell types (Fig. 8C). We further analyzed at different times the instantaneous receptive field (IRF) of ganglion cells, which is the gradient of a cell's response with respect to the stimulus (Fig. 8B,D), revealing the visual feature driving the cell for a particular stimulus. We found that single modes could contribute similar IRF patterns across different ganglion cells. Interestingly, when multiple modes simultaneously drove a cell, the resulting IRF dynamically varied according to the combination of active modes, with patterns ranging from familiar center-surround structures to oriented or textured responses (Fig. 8D). Previous studies have indicated that groups of ganglion cells can be driven by visual features that are the intersection of their receptive fields (Schnitzer and Meister (2003)), and that decomposition of ganglion cell activity can reveal modes with error-correcting properties that represent localized features distinct from those of single neurons (Prentice et al. (2016)). As the units of this model are highly correlated with actual interneuron recordings (Maheswaranathan et al. (2023)), contribution modes we identify here serve to automatically generate experimentally testable hypotheses that the presynaptic neurons with the visual features identified by the model drive ganglion cell activity patterns with error correcting properties.

## 8 CODEC ON VISION TRANSFORMERS

We additionally applied CODEC to three layer types—token features, MLPs, and attention layers—across the depth of a standard Vision Transformer, ViT-B (Dosovitskiy et al. (2020)) (see Supp. Sec. 10.5). Drawing an analogy to CNNs, we took a straightforward approach by treating tokens as the spatial dimension and hidden dimensions as channels. As in CNNs, we found that contributions are sparser than activations. Subsequent decomposition revealed distinct contribution modes for these three layer types across all layers. These modes could be leveraged for targeted perturbation experiments more effectively than those based on activations, revealing causal information not cap-

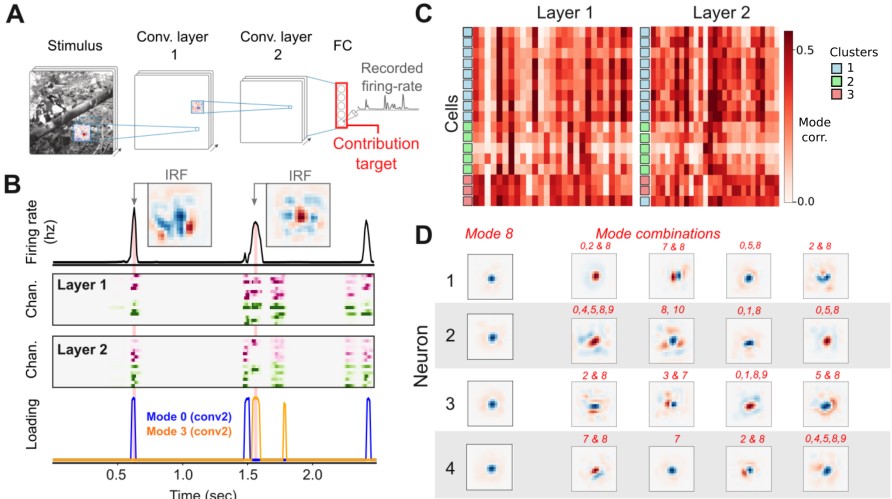

Figure 8: **Contribution modes generate dynamic receptive fields in the retina.** (A) CNN trained to predict retinal ganglion cell responses to natural stimuli, from Ding et al. (2023). CODEC is performed for hidden units to single target outputs including surprisal (self-information) of the population or from single cell responses. (B) Model firing rate predictions for an example cell (top) aligned with the matrix of contributions, and with loadings for two example modes after SAE decomposition (bottom). Also shown are instantaneous receptive fields (IRFs) of the cell computed at two time points. (C) Clustering of cells using their contribution modes. Matrix shows the correlation of firing of each cell with the loading of each mode. Three clusters (number selected using silhouette value) computed using $k$-means remain consistent across layers. (D) IRFs during sparse mode combinations. Left column: IRFs when one specific mode was active alone for four different cells at different times. Right: IRFs for example cells at times when 5 or fewer modes were active and cells were firing more than 1 Hz. Each row is from a different cell.

tured by activations alone. Although overall ablation performance was inferior to that observed in CNNs, we note that ViTs utilize fundamentally different computational strategies due to their lack of an explicit spatial equivariant bias. Consequently, further exploration is needed to determine the optimal spatial reduction strategy for ViTs, beyond our initial approach of treating tokens as space.

## 9 CONCLUSION

Contribution decomposition identifies how hidden units construct specific outputs, revealing both the input components that causally drive model behavior and how the effects of those features are summed across outputs. Our approach thus achieves both a deeper understanding of the computation than results from examining representations alone, along with more effective manipulation of those networks. Our analysis reveals insights into the structure of neural computation, in particular at intermediate layers of networks that have been difficult to analyze and interpret. In biological networks such as the retina, CODEC allows an analysis of how dynamic sensitivity to visual input arises from the coordinated actions of model interneurons. In artificial networks, the emergence of sparse, interpretable motifs suggests that network output can be understood in terms of a relatively small set of input-specific computations. Future work might leverage these computations as building blocks for more efficient architectures or transfer learning approaches. In the nervous system, the ability to relate computational building blocks to a downstream computation, combined with experimental measurements and manipulations at different levels could reveal how information is recombined across diverging and converging neural pathways. Such a unified approach to neural computation could bridge the gap between computers and biology, potentially enabling more powerful AI systems and deeper insights into biological intelligence.

LIMITATIONS AND ETHICAL CONCERNS

Whereas we focused on image classification, our approach lays the groundwork for extensions to other architectures and applications. Some analyses on ResNet-50 were limited to specific blocks rather than the entire network, and although CODEC is architecturally agnostic and could be applied to interpret more complex models such as LLMs, we have at this point only analyzed scaling with LLMs (see supplemental material).

REPRODUCIBILITY STATEMENT

The key aspects of the CODEC framework are implemented as a publicly available repository, containing scripts to reproduce each figure of this manuscript, available at **https://github.com/baccuslab/CODEC_ICLR_2026**. Here, we also provide the sparse autoencoder decomposition algorithms and visualization tools for mapping contributions back to input space, as additionally documented in the supplemental material, with mathematical derivations provided. The retinal neural network models analyzed are publicly available on GitHub as referenced in the citations. All experimental configurations, including autoencoder architectures, sparsity constraints, and statistical analysis procedures, are documented to enable direct replication of our findings across both artificial and biological network interpretations.

ACKNOWLEDGEMENTS

This work was supported by grants from the National Eye Institute (NEI): R01EY022933, R01EY025087 and P30EY026877 (SAB). S.G. thanks the Simons Foundation Collaboration on the Physics of Learning and Neural Computation and a Schmidt Sciences Polymath Award for support. Z.A. was supported by the Stanford Medicine Post-Baccalaureate Experience In Research program.

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

## 10 SUPPLEMENTAL MATERIAL

### 10.1 SPATIAL AGGREGATION AND E/I SEPARATION

Once the contributions have been computed using the chosen contribution method and target, we obtain a map of the intermediate layer of the same size as the layer's activations. For convolutional layers, we perform spatial summation over the height and width dimensions to derive a single contribution value per channel:

$$\text{ChanC} = \sum_{h=1}^{H} \sum_{w=1}^{W} \text{IG}_{:,h,w}, \tag{1}$$

where "ChanC" means "channel contribution". We further reduce these contributions into excitatory (positive) and inhibitory (negative) components:

$$\text{ChanC}^{+} = \max(\text{ChanC}, 0); \quad \text{ChanC}^{-} = \min(\text{ChanC}, 0). \tag{2}$$

This E/I decomposition enhances interpretability by separating units that promote versus suppress specific outputs, revealing antagonistic computational mechanisms within the network. For analyses of modes presented here, we focus exclusively on the positive contributions, which can be interpreted as the hidden unit contributions that positively drive the attribution target. However, future work could extend this approach to also incorporate negative contributions, depending on the nature of the output target.

### 10.2 CORRELATION ANALYSIS WITH SEMANTIC CONCEPTS

To interpret the discovered modes, we analyzed their correlation with known semantic concepts from the ImageNet hierarchy. We constructed binary masks $\mathbf{M} \in \{0,1\}^{n \times c}$ representing $c$ different semantic concepts , where $M_{i,j} = 1$ if sample $i$ belongs to a higher order concept/ ImageNet Class $j$ and 0 otherwise.

We then computed the Pearson correlation coefficient between each mode loading and each concept mask:

$$\text{Corr}(L_i, M_j) = \frac{\text{Cov}(L_i, M_j)}{\sigma_{L_i} \sigma_{M_j}} \tag{3}$$

where $L_i$ is the $i$-th column of the loading matrix $\mathbf{L}$ (corresponding to mode $i$), $M_j$ is the $j$-th column of the mask matrix $\mathbf{M}$ (corresponding to concept $j$), Cov is the covariance, and $\sigma$ is the standard deviation.

This analysis enabled us to assign interpretable meanings to the learned modes and understand how semantic information is distributed across the network.

### 10.3 ROBUSTNESS TO SAE HYPERPARAMETERS

To examine the stability and sensitivity of sparse autoencoders (SAE) modes trained on contributions as a function of hyperparameters, we trained SAEs on ResNet-50 intermediate layer contributions on example layers 3, 7, 13 and 15 while varying six hyperparameters: L1 sparsity penalty strength, threshold, dictionary size (N * number of channels), random seed, MLP encoder size, and a non-negativity constraint (Fig. S3). For each hyperparameter variation, we computed the correlation between the learned modes and a baseline SAE configuration (used in the main text) by aggregating modes for each ImageNet class—summing all modes with correlation $> 0.2$ to that class, or using the single top-correlated mode if none exceeded this threshold. Varying the L1 sparsity penalty (0, 1e-04, 1e-02) showed minimal impact on learned representations. The threshold for mode selection (0.5, 0.7, 0.9) produced moderate variation, particularly in middle layers when the threshold was not restrictive enough. The dictionary size multiplier N (×1, ×3, ×5 the layer channel dimension) indicated that smaller dictionaries (N=1) produce less stable representations in later layers, indicating the initial dictionary size should be set greater than number of channels in the layee. Random seed variation demonstrated high reproducibility with near-identical correlations to baseline SAE across all layers. Varying the MLP hidden dimension (128, 512, 2048, 4096) indicated a minimum size

was needed, as smaller hidden dimensions created a bottleneck for layers with higher input dimensionality, degrading reconstruction performance. Non-negative constraint on modes shows moderate effects on mode correlation to baseline, with the positive constraint producing slightly higher correlation in deeper layers. All SAEs indicate that later layers learn sparser, more distributed class representations.

We further examined performance metrics of SAEs as a function of hyperparameters. Reconstruction of raw contributions was consistently high (0.7–0.95) across all hyperparameters (Fig. S4. For modes with correlation to image class $> 0.2$, the number of class-correlated modes increased with layer depth. Larger dictionary sizes (N=5) yielded a greater number of class-correlated modes, however the sum of those modes showed high class correlation insensitive to dictionary size, suggesting that additional modes capture redundant rather than distinct information. Very large MLP encoders yielded fewer class correlated modes, likely due to feature splitting (Bricken et al. (2023)). Conversely, the number of classes having a correlation with modes of $> 0.2$ also was insensitive to variation of L1 strength, random seed, MLP size, and constraint of MLP sign. As with other measures, the dictionary size multiplier (N) needed to be greater than 1, suggesting that sufficient initial overcompleteness is necessary for the SAE to discover class-specific modes.

With respect to specific classes, changing hyperparameters had minimal effect on SAE performance and learned representations. We summed all modes with correlation $> 0.2$ for the black widow class in layer 15 (Fig. S5), finding that despite different configurations producing varying numbers of correlated modes, the summed representations remained strikingly similar.

Systematic hyperparameter sweeps with all other parameters held constant revealed robust specificity patterns: dictionary size multiplier (N=1–5), activation threshold (0.5–0.9), mode L1 regularization (0–1e-2), random seed, and MLP hidden layer size (128–4096) all produced comparable ablation specificity with consistent trends of increased specificity in deeper layers, demonstrating that our interpretability results generalize across architectural choices (Fig. S6).

### 10.4 RUNTIME MEASUREMENTS AND COMPLEXITY

We measured contribution computation time for ResNet-50 across all 16 layers on the ImageNet validation set (50,000 images) on an NVIDIA RTX 6000 GPU. Using integrated gradients with 10 interpolation steps takes approximately 4.1 hours total, averaging 4.7 seconds per batch of 16 images (293 ms per image). In contrast, activation $\times$ normalized gradient (act_normgrad) completes in 1.3 hours, averaging 1.5 seconds per batch (93ms per image).

We report runtime measurements for training SAEs over 200 epochs on an NVIDIA RTX 6000 GPU. For Resnet-50, SAEs for earlier layers train relatively quickly due to having fewer channels, with Layer 3 completing in $\sim$ 3 min and Layer 15 in $\sim$ 7 minutes. For ViT, training takes $\sim$ 5 minutes per layer for attention and token features, and $\sim$ 30 minutes for MLP (it has size 3072 compared to 768 for the other two layer types).

Computing contributions with activation $\times$ gradient is $O(P)$ where $P$ is the number of model parameters, requiring one forward pass to obtain activations and one backward pass to compute gradients with respect to the target objective. Activation $\times$ integrated gradients has complexity $O(n_{steps} \cdot P)$, scaling linearly with both the number of parameters and interpolation steps. In our experiments with $n_{steps} = 10$, this translated to approximately 3$\times$ compute time compared to activation $\times$ gradient.

Once contributions are precomputed, training contribution-SAEs has the same computational cost as training activation-SAEs, operating on feature vectors of identical dimensionality. This makes the contribution computation method the primary bottleneck, with activation $\times$ gradient enabling practical large-scale analysis while Integrated Gradients provides more theoretically grounded contributions at $n_{steps}\times$ higher cost. The choice of decomposition method (SAE, PCA, or analyzing raw contributions) depends on the scientific question and represents a separate complexity consideration.

To explore the scaling of CODEC to larger models, we additionally benchmarked contribution computation on Gemma-3n (5.44B parameters, 30 layers) across multiple NVIDIA Titan XP GPUs. Contributions were computed with respect to output entropy. At a sequence length of 128 tokens, activation $\times$ normalized gradient completes in approximately 720ms per input, while integrated gradients (10 steps) requires approximately 8 seconds. Runtime remains invariant to hook location (*post-attention* vs. *post-feedforward*) and layer depth (early, mid, late), and scales sub-linearly with

sequence length (empirical scaling exponent $\alpha \approx 0.10\text{–}0.11$), projecting to approximately 1 second (ActGrad) and 11 seconds (IntGrad) even at 4096-token contexts. These results confirm that CODEC's contribution computation generalizes beyond vision models to large-scale language architectures. We note that SAE training is a separate computational step, independent of contribution computation, and scaling it to LLM-scale features is left to future work.

## 10.5 CODEC ON VITS

CODEC is a general framework that can be applied to any feedforward neural network. To show its application across architectures, we apply CODEC to ViT-B, which is an encoder-only transformer with 12 transformer blocks (Fig. S7). Because ViTs do not have the same channel and layer structure as CNNs, we log activations and gradients to compute contributions at three types of layers for each transformer block—token features, MLP, and attention—and make the appropriate summation that parallels the spatial sum we perform for each channel in CNNs. Below are the details of implementation:

- Token features: We log the activations and gradients at the end of each transformer block after the merge into the residual stream, with shape (batch, num_tokens, hidden_dim).
- MLP: We log the activations and gradients at the hidden layer of the MLP, after the nonlinearity, with shape (batch, num_tokens, MLP_hidden_dim).
- Attention: We log the activations and gradients after self attention but before the out projection (mixing), with shape (batch, num_tokens, num_heads * head_dim).

We use Integrated Gradients with 10 integration steps as the contribution algorithm for the following results. We use the post-softmax logit corresponding to the top predicted target class as the contribution target. After we compute the contributions for each neuron, we perform the appropriate summation to obtain a tensor of shape (batch, pseudo-channels):

- Token features: Because each token is a patch, we sum over the tokens to obtain a tensor of shape (batch, hidden_dim), where the hidden dimension corresponds to channels, intuitively interpreted as "global feature detectors".
- MLP: We treat each MLP hidden neuron as its own channel, so we sum over tokens to obtain a tensor of shape (batch, MLP_hidden_dim).
- Attention: We treat each hidden dimension in each attention head as a channel, so we sum over both tokens to obtain a tensor of shape (batch, num_heads * head_dim).

For each of the three layer types in ViT, we compute a contribution tensor analogous to that of CNNs—shape (batch, channels)—and we term the second dimension pseudo-channels. Like CNNs, we separate positive and negative contributions within each pseudo-channel and concatenate them to obtain (batch, 2*channels) before the summation, where half of the channels only contain the positive contributions with the negative contributions set to zero and vice versa. Then, we take the positive channels and use SAEs to find contribution modes, interpreted as groups of channels acting together to positively drive the output. For activations, we do not separate into positive and negative components before the SAE because the sign does not carry special meaning for activations as opposed to contributions (where a positive contribution predicts a decrease in the contribution target if ablated). We scanned minimally for SAE hyperparameters and settled on the default hyperparameters used for ResNet-50.

A note about completeness: Because MLP and attention contributions are computed within a transformer block rather than on the full residual stream, their contributions will not be complete. Intuitively, even if one ablates the entire MLP or all attention heads, the residual stream still carries over information from the previous block, which can still maintain good classification performance. Therefore, of the three layer types, we should only expect token features ablations to converge to zero performance when all pseudo-channels are ablated. In fact, empirically, we observe that the ablation of an entire MLP or attention layer within a single transformer block only results in a $\sim 10\%$ decrease in performance.

We note that ViTs could be utilizing very different computational strategies than CNNs for image classification. In particular, there is no explicit spatially equivariant bias in the transformer architecture. We applied CODEC to ViTs in a straightforward manner analogous to CNNs by summing over

"space" (tokens) to obtain spatially invariant feature detectors or information streams, which may not be how ViTs organize computation in its dimensions due to the lack of this bias. Therefore, we should not expect the same level of general perturbation performance as CNNs for the same SAE architecture and hyperparameters. However, we show below that contributions as a causal measure still reveal causal information not captured by activations.

### 10.5.1 SPARSITY

Plotting the contributions and activations for a single image at the three layer types at layer 9, we see that contributions provide a sparser description of the model (Fig. S7A–C). Using the same measures of sparsity as in Resnet-50, we computed the average sparsity (over images) for the three layer types across the entire network (Fig. S7D–F). As in Resnet-50, contributions are more consistently sparse than activations. Additionally, we observe similarities and differences in sparsity trends for the three layer types across depth. For all three layer types, we observe the general "inverted U" trend, where sparsity goes down at the beginning and eventually increases towards the deep end of the model. However, for MLP we additionally observe near maximum sparsity approaching the final layers but a drop in sparsity at the final layer. These observations suggest distinct sparse organizations used by the different layer types, the full exploration of which we leave for future work.

### 10.5.2 CORRELATION ANALYSIS

We compute the average (across images) maximal correlation to target classes for four units— activations of pseudo-channels, contributions of pseudo-channels, activation modes, and contribution modes. Shown in Fig. S7G–I are these correlations for the three types of layers that we analyze.

Across layer types, we observe an increase in correlations as depth increases, except the slight drop for MLP. Furthermore, we observe that modes are more correlated with target classes than individual pseudo-channels for token features and attention, with activation modes more correlated than contribution modes. This suggests that as depth increases, the model not only increasingly represents but also drives target classification with its hidden units. It also increasingly uses groups of hidden dimensions to collectively represent or drive the output classification.

### 10.5.3 PERTURBATION ANALYSIS

We also performed ablation analyses on ViT-B using CODEC, probing the causal efficacy of contribution-SAE against activation-SAE, similar to Fig. 6. For each layer and a specified percentage of pseudo-channels to ablate, we take six trials as follows. In each trial, we take 10 random target classes. For each target class, we first find the most correlated activation or contribution mode. Then, we randomly select a class that is not the target class ("off-target" class). We take 10 samples from each class. Then, we ablated a range of specified percentages (from 1% to 99%) of the most prominent channels in the mode and evaluated performance ratio of the perturbed model to the original model on the on-target and off-target classes using the 10 images that we just sampled for each class. The performance ratio was then averaged across the 10 target classes as one measurement of the performance of that layer across ablation percentages. The area under this performance ratio-ablation percentage curve (AUC) was computed for on-target and off-target ablations and the difference taken to quantify ablation specificity. The AUC ranges from 0 (performance drops to 0 as soon as the first channel is ablated and stays there) to 1 (performance stays at original until all channels are ablated) except for rare cases where perturbed performance exceeds original performance. We plot the 6 AUC differences for each layer and also show the mean and SEM for each layer in Fig. S7J–L. We summarize findings below:

**Token features.** Contributions yield significant ablation specificity, since the AUC for on-target ablations is significantly below that of off-target ablations. Contributions are also significantly more effective than activations, especially at late layers. Furthermore, as depth increases into the last layers, contribution performance mostly monotonically increases while activation performance stays mostly flat and similar to the baseline before deteriorating into the last layers, even though correlations of both contributons and activations increase monotonically. For activations, ablating their modes in the deeper half of the model actually hurts off-target classes more than on-target classes, suggesting a unique computational strategy used by ViTs to represent classes in hidden units that

causally oppose those classes. This organization may indicate an inhibitory effect similar to inhibitory neurons that carry information about a specific visual feature but suppress that same feature for gain control or spatial decorrelation (Ölveczky et al. (2003)). Even though class information is better represented by activations as depth increases, the causally contributing components that drive classification drift away from the most prominently activating token features (which become causally inhibitory), necessitating the use of a causal measure like CODEC to capture the causal effects in the late layers.

**MLPs.** The overall ablation effect for MLPs is weaker than for token features, which is expected because MLP neurons do not include the residual stream. However, contribution-SAE still yields significant ablation specificity for later layers compared to baseline and is significantly more effective than activation-SAE. Because ablating the entire MLP in one block usually only yields a $\sim 10\%$ decrease in performance (which would translate to a -0.1 floor of the AUC difference), both contribution-SAE and activation-SAE ablations approach this floor at layer 10, but contribution-SAE ablations are more effective in the layers leading up to 10. Furthermore, we observe a sharp performance drop to baseline at the last layers for both, despite the fact that correlations for both contribution and activation modes stay high into the last layer. One possibility for this effect is that the most causally relevant components may be shifted into the residual stream as depth increases, and what is remaining in the MLP at these last layers more clearly represent the target classes but are less causal.

**Attention.** The overall ablation effect for MLPs is even weaker than for MLPs, but contribution-SAE still yields specificity at the deepest layers and is significantly more effective than activation-SAE, which stays flat near the baseline. Compared to the -0.1 maximum possible performance, contribution-SAE gets roughly halfway there in the last few layers.

Overall, a straightforward application of CODEC on ViTs by treating hidden dimensions as channels and tokens as spatial patches shows that contribution-SAE enables target class ablations better than activation-SAE. Even though ablation performance is weaker than in CNNs (the comparable case being token features, which are taken after the merge into the residual stream), we note that summing over tokens as space may not align with how ViTs organize computation because of their lack of explicit spatially equivariant bias. We leave an exploration of the optimal spatial reduction strategy for ViTs to future work.

### 10.5.4 ATTENTION HEAD SPARSITY

Unlike ResNet-50, ViTs utilize multihead attention. We apply CODEC to investigate whether individual attention heads specialize in functions that causally drive output classification.

We analyzed head specialization through contribution sparsity, which can manifest at the macro scale (few highly active heads) or the micro scale (active dimensions scattered evenly across all heads). To quantify this, we compute the Hoyer sparsity index on a 12-dimensional vector—derived by summing contributions or activations within each head—and compare it against a baseline of randomly grouped hidden dimensions. A sparsity index higher than the baseline indicates head specialization.

As shown in Fig. S8, randomly grouped heads show nearly identical sparsity for contributions and activations. Learned attention heads show a small increase in sparsity for both contributions and activations, but the increase is well within the fluctuation over the dataset. Recall from Fig. S7F that contributions are about 0.1 to 0.2 higher in sparsity than activations for attention across depth, and sometimes exceed 0.5 sparsity. Comparatively, the head-level sparsity is much smaller than the sparsity observed at the individual channel level. Therefore, these results demonstrate that attention heads are minimally specialized in representations as well as causal functions, with active and contributing hidden dimensions spanning across attention heads.

### 10.6 CONTRASTIVE TARGETS

In addition to the targets listed in the main text, we also computed contributions with respect to a contrastive target (Fig. S11). Both contrastive and top-1 targets produced similarly sparse representations across all network layers, demonstrating that the contrastive target yields sparsity patterns

consistent with the top-1 approach used in the main paper. We successfully decomposed contrastive contributions with $R^2$ above 0.85 for the four analyzed layers, suggesting that contrastive targets can be used for similar causal analysis.

### 10.7   CODEC IN DEPTH: ADDITIONAL DETAILS ON CONTRIBUTION ALGORITHMS AND THEIR COMPLETE DECOMPOSITIONS IN INPUT SPACE

The fundamental algorithmic innovation of our work is that we design an approach to address the question of how combinations of hidden neurons drive network output. This contrasts with an analysis of activations, which at any level and for any input are not guaranteed to drive network output. Here, we provide additional details of our methods.

We revisit and describe the motivation behind our several contribution algorithms, which are inspired by gradient-based attribution methods, and we derive their complete decompositions in input space. For notation, let $h_i$ be the hidden units in a specific layer $L$. All attribution methods need a scalar output target, typically a select output neuron or a scalar function of the output neurons. Let $T$ denote the target (for example the output logit of a target class) and $y := f_T$ be the neural network with a scalar output $y$ corresponding to $T$. For clarity, we define the complete input space decomposition of hidden units:

**Definition 1** (Complete input space decomposition of hidden units)**.** *Let $C_j$ be the contribution of hidden unit $j$ and $\widetilde{C}_{ij}$ be an input space decomposition of $C_j$, i.e., $\widetilde{C}_{ij}$ is the part of $C_j$ assigned to input pixel $i$. Then, we call this input space decomposition complete if*

$$\sum_i \widetilde{C}_{ij} = C_j. \tag{4}$$

We derive the complete input space decomposition of ActGrad, InputGrad, and Integrated Gradients under this definition, which will naturally generalize to mode contributions by linearity as a simple sum over the all the hidden units in that mode.

**A note about ActGrad**   To be consistent with the rest of the derivation, we slightly generalize ActGrad to account for possible nonzero baselines:

$$\text{ActGrad}_j = (h_j - h'_j) \cdot \frac{\partial y}{\partial h_j}, \tag{5}$$

where $h_j$ is the activation of hidden unit $j$ (which could be the input as a special case) at input $\mathbf{x}$ and $h'_j$ is the hidden activation at baseline input $\mathbf{x}'$. For image classification, we usually take a baseline hidden activation of zero, which is the definition in the main text.

**Input $\times$ Gradient (InputGrad)**   Used in Shrikumar et al. (2017) as a baseline, this is a special case of ActGrad where Act is the input values. However, seeing InputGrad this way means it can only attribute to the input layer. We naturally extend InputGrad to hidden layers to obtain contributions by first defining the input space decomposition:

$$\widetilde{\text{InputGrad}}_{ij} := \frac{\partial y}{\partial h_j} \frac{\partial h_j}{\partial x_i} (x_i - x'_i) \tag{6}$$

$$= ((J_y)_j (J_z)_{ji}) \cdot (x_i - x'_i), \tag{7}$$

where $J_y := \frac{\partial y}{\partial h_j}$ is the Jacobian from the output to the hidden layer and $J_h := \frac{\partial h_j}{\partial x_i}$ is the Jacobian from the hidden layer to the input. If we sum over $i$, we then obtain a contribution algorithm to hidden neurons:

$$\text{HInputGrad}_j := \sum_i \widetilde{\text{InputGrad}}_{ij}. \tag{8}$$

Under this definition of InputGrad contribution, the input space decomposition is trivially complete. This is a well-motivated extension of InputGrad because we recover the original InputGrad by summing over $j$ instead:

$$\text{InputGrad}_i = \sum_j \widetilde{\text{InputGrad}}_{ij}. \tag{9}$$

This natural extension rests on the chain rule of partial derivatives, where we essentially postpone summing gradients over the hidden layer until after multiplying by the input element-wise. InputGrad as a contribution algorithm may seem a bit ad hoc, but it will turn out to have the most computationally efficient input space decomposition that we consider and we theoretically motivate it later by showing a connection to Integrated Gradients.

**Integrated Gradients (IG) (Sundararajan et al. (2017))** IG assigns attributions to input pixels by calculating the integral of gradients along a straight-line path from a baseline input $\mathbf{x}'$ to the actual input $\mathbf{x}$:

$$\text{IG}_i = (x_i - x_i') \times \int_{\alpha=0}^{1} \frac{\partial y(\mathbf{x}' + \alpha(\mathbf{x} - \mathbf{x}'))}{\partial x_i} d\alpha. \tag{10}$$

IG satisfies the (output) *completeness* property, often desired in attribution methods:

$$\sum_i \text{IG}_i = y(x) - y(x'). \tag{11}$$

In words, completeness means that the attributions to each input pixel sum up to the (change in) output. In this original form, IG only works for the input layer. We present two ways to extend IG to hidden layers:

- Treat the hidden layer as the input and follow through. This is the extension by Lucas et al. (2022), where they combine this extension of IG with GradCAM. Mathematically, the "hidden integrated gradients" (HIG) is defined as

$$\text{HIG}_j = (h_j - h_j') \times \int_{\alpha=0}^{1} \frac{\partial y(\mathbf{h}' + \alpha(\mathbf{h} - \mathbf{h}'))}{\partial h_j} d\alpha, \tag{12}$$

  where $\mathbf{h}$ is the hidden layer activation vector at input $\mathbf{x}$, and $\mathbf{h}'$ is the hidden layer activation vector at input $\mathbf{x}'$.

- Take the same path in input space but decompose the gradients at the hidden layer and sum over the input layer. Tanaka et al. (2019) first introduced this method and applied it to the first hidden layer of a model of the retina. Here, we use it on any hidden layer. Mathematically, first, we decompose the gradients over the hidden layer:

$$\widetilde{\text{HIG}}_{ij} := (x_i - x_i') \times \int_{\alpha=0}^{1} \frac{\partial y(\mathbf{x}' + \alpha(\mathbf{x} - \mathbf{x}'))}{\partial h_j} \frac{\partial h_j}{\partial x_i} d\alpha, \tag{13}$$

  where $i$ indexes the input pixels and $j$ indexes the hidden neurons. Note that we can recover standard IG by summing over $j$ due to the linearity of the integral and chain rule of partial derivatives:

$$\text{IG}_i = \sum_j \widetilde{\text{HIG}}_{ij}, \tag{14}$$

  which motivates this extension. Now, to obtain contributions in the hidden layer, we simply sum over $i$ instead of $j$:

$$\text{HIG}_j := \sum_i \widetilde{\text{HIG}}_{ij}. \tag{15}$$

In practice, we approximate the integral using a Riemann sum with $m$ steps. For standard IG, we have

$$\text{IG}_i \approx (x_i - x_i') \times \frac{1}{m} \sum_{k=1}^{m} \frac{\partial y(\mathbf{x}' + \frac{k}{m}(\mathbf{x} - \mathbf{x}'))}{\partial x_i}. \tag{16}$$

This approximation extends straightforwardly to the first method of HIG, simply by replacing $\mathbf{x}$ with $\mathbf{h}$. For the second method, we could naively discretize in input space to get $\widetilde{\text{HIG}}_{ij}$ first, but that would involve a backward pass for each hidden neuron at each step. Instead, we analytically derive

an approximation that still only involves one backward pass at each step. We have

$$\text{HIG}_j = \sum_i (x_i - x_i') \times \int_{\alpha=0}^1 \frac{\partial y(\mathbf{x}' + \alpha(\mathbf{x} - \mathbf{x}'))}{\partial h_j} \frac{\partial h_j}{\partial x_i} d\alpha \tag{17}$$

$$= \int_{\alpha=0}^1 \sum_i \left[ \frac{\partial y}{\partial h_j} \frac{\partial h_j}{\partial x_i} \frac{\partial x_i}{\partial \alpha} \right]_{\mathbf{x}' + \alpha(\mathbf{x} - \mathbf{x}')} d\alpha \tag{18}$$

$$= \int_{\alpha=0}^1 \left[ \frac{\partial y}{\partial h_j} \sum_i \frac{\partial h_j}{\partial x_i} \frac{\partial x_i}{\partial \alpha} \right]_{\mathbf{x}' + \alpha(\mathbf{x} - \mathbf{x}')} d\alpha \tag{19}$$

$$= \int_{\alpha=0}^1 \left[ \frac{\partial y}{\partial h_j} \frac{\partial h_j}{\partial \alpha} \right]_{h(\mathbf{x}' + \alpha(\mathbf{x} - \mathbf{x}'))} d\alpha. \tag{20}$$

Note that this is precisely IG treating $\mathbf{h}$ as the input layer (the first extension) with the important distinction that the path is not a straight line (it is warped by the nonlinear functional relationship between $\mathbf{x}$ and $\mathbf{h}$). The crucial advantage of computing HIG this way over the first method is the complete input space decomposition (Equation 13) we get for free. To approximate this line integral with a Riemann sum, we have

$$\text{HIG}_j \approx \sum_{k=1}^m \frac{\partial y}{\partial h_j} \bigg|_{\mathbf{h}(\alpha = \frac{k}{m})} dh_j, \tag{21}$$

where we can compute $\mathbf{h}(\alpha = \frac{k}{m})$ at each $k$ in a forward pass and $dh_j = h_j(\alpha = \frac{k}{m}) - h_j(\alpha = \frac{k-1}{m})$ in a single backward pass.

**Complete input space decomposition of ActGrad**   It is a bit more nuanced to define a complete input space decomposition for ActGrad in hidden layers. The idea is to use IG to attribute Act to inputs, which we then multiply by Grad to get the decomposition. Mathematically, we exploit the output completeness of IG to express Act as an integral of gradients, which we can then naturally decompose linearly in input space. We define the input space decomposition as

$$\widetilde{\text{ActGrad}}_{ij} := (x_i - x_i') \frac{\partial y}{\partial h_j} \bigg|_{\mathbf{x}} \int_{\alpha=0}^1 \frac{\partial h_j}{\partial x_i} d\alpha, \tag{22}$$

where we take a straight-line path in input space parameterized by $\alpha$. We can prove that this decomposition is complete by recovering the standard ActGrad for hidden layers if we sum over $i$:

$$\sum_i \widetilde{\text{ActGrad}}_{ij} = \sum_i (x_i - x_i') \frac{\partial y}{\partial h_j} \bigg|_{\mathbf{x}} \int_{\alpha=0}^1 \frac{\partial h_j}{\partial x_i} d\alpha \tag{23}$$

$$= \frac{\partial y}{\partial h_j} \bigg|_{\mathbf{x}} \sum_i (x_i - x_i') \int_{\alpha=0}^1 \frac{\partial h_j}{\partial x_i} d\alpha \tag{24}$$

$$= \frac{\partial y}{\partial h_j} \bigg|_{\mathbf{x}} (h_j - h_j') \quad \text{(due to the output completeness of IG)} \tag{25}$$

$$= \text{ActGrad}_j, \tag{26}$$

where the second-to-last step is true even though the path in $h$ space is not necessarily straight by the fundamental theorem of calculus for line integrals. Note that if we summed over $j$ instead, we would obtain a new attribution method for the input layer that is like a hybrid between IG and InputGrad (the difference being which part of the chained derivative is integrated over). We do not explore that method any further as we are primarily interested in contributions of hidden neurons.

**InputGrad and ActGrad as approximations to IG**   The unifying connection behind all three contribution algorithms is that they are the same for a linear network. The three algorithms represent successively more comprehensive ways to capture the nonlinearity of the network in their contribution assignments. Recall the definition of HIG (Equation 20):

$$\text{HIG}_j = \int_{\alpha=0}^1 \left[ \frac{\partial y}{\partial h_j} \frac{\partial h_j}{\partial \alpha} \right]_{h(\mathbf{x}' + \alpha(\mathbf{x} - \mathbf{x}'))} d\alpha. \tag{27}$$

If we assume that the downstream network is linear along the integration path, i.e., the downstream gradients $\frac{\partial y}{\partial h_j}$ are constant, we recover ActGrad:

$$\int_{\alpha=0}^{1} \left[ \frac{\partial y}{\partial h_j} \frac{\partial h_j}{\partial \alpha} \right]_{h(\mathbf{x}'+\alpha(\mathbf{x}-\mathbf{x}'))} d\alpha = \left. \frac{\partial y}{\partial h_j} \right|_{\mathbf{x}} \int_{\alpha=0}^{1} \left[ \frac{\partial h_j}{\partial \alpha} \right]_{h(\mathbf{x}'+\alpha(\mathbf{x}-\mathbf{x}'))} d\alpha \tag{28}$$

$$= \left. \frac{\partial y}{\partial h_j} \right|_{\mathbf{x}} (h_j - h_j') \tag{29}$$

$$= \text{ActGrad}_j. \tag{30}$$

If we further assume the whole network is linear along the integration path, i.e., all gradients are constant, we recover InputGrad:

$$\int_{\alpha=0}^{1} \left[ \frac{\partial y}{\partial h_j} \frac{\partial h_j}{\partial \alpha} \right]_{h(\mathbf{x}'+\alpha(\mathbf{x}-\mathbf{x}'))} d\alpha = \int_{\alpha=0}^{1} \left[ \frac{\partial y}{\partial h_j} \sum_i \frac{\partial h_j}{\partial x_i} \frac{\partial x_i}{\partial \alpha} \right]_{\mathbf{x}'+\alpha(\mathbf{x}-\mathbf{x}')} d\alpha \tag{31}$$

$$= \left[ \frac{\partial y}{\partial h_j} \sum_i \frac{\partial h_j}{\partial x_i} \frac{\partial x_i}{\partial \alpha} \right]_{\mathbf{x}} \int_{\alpha=0}^{1} d\alpha \tag{32}$$

$$= \sum_i \frac{\partial y}{\partial h_j} \frac{\partial h_j}{\partial x_i} (x_i - x_i') \tag{33}$$

$$= \sum_i \widetilde{\text{InputGrad}}_{ij} \tag{34}$$

$$= \text{InputGrad}_j. \tag{35}$$

By taking the full integral, IG has the output completeness property $\sum_j \text{HIG}_j = y - y'$ that ActGrad and InputGrad don't have. Furthermore, IG alleviates the saturation problem where a zero gradient counterintuitively leads to a zero attribution (and therefore contribution) (Shrikumar et al. (2017); Srinivas and Fleuret (2019)).

Under our definition, a mode's contribution is the sum of the contributions of its top-$k$ channels. Due to the linearity of sums and integrals, the complete input space decompositions derived above for individual hidden units naturally generalize to those of modes.

The key insight that underlies input space completeness is the chain rule of partial derivatives and linearity of integrals and sums (and additionally the output completeness of IG for ActGrad decomposition). The high-level procedure is:

1. When computing the contribution of each hidden unit, do not sum the gradients over the input space just yet; save the full contribution tensor with the input index.

2. Sum over the input space and cluster to find modes.

3. Go back to the unsummed contributions and sum over each *mode* to get a complete decomposition of mode contributions in input space.

Practically, it is more computationally expensive to use the input space decomposition of IG and ActGrad than InputGrad due to the integration. Therefore, we use InputGrad as the backbone algorithm for all our input space visualizations. Anecdotally, for attributions to input space, Selvaraju et al. (2020) notes that ActGrad at earlier layers produce less clear heatmaps, but since we use IG at the hidden layer first to identify modes, we do not expect the loss in quality to be as significant by switching to InputGrad just for visualization in input space. Empirically, our experimental visualizations show sufficient quality for our purposes. However, our theoretical framework allows for a principled, complete decomposition if necessary for the application at hand.

## 10.8 VISUALIZING CONTRIBUTION MAPS

For each image, we computed contribution maps using the InputGrad method described in the main text. Specifically, for a target class and a set of mode-selected channels identified via CODEC analysis, we computed the Jacobians $J_{y,h_{c,p}} = \partial y / \partial h_{c,p}$ (gradient of the softmax class output

with respect to each hidden unit activation) and $J_{h_{c,p},x_i} = \partial h_{c,p}/\partial x_i$ (gradient of each hidden unit activation with respect to each input pixel), retaining spatial resolution across all positions $p$ in the feature map. The per-channel, per-position input sensitivity was then formed as $A_i^{(c,p)} = J_{y,h_{c,p}} \cdot J_{h_{c,p},x_i}$, and the contribution map was computed as the elementwise product $C_i^{(c,p)} = A_i^{(c,p)} \odot x_i$. Contributions were aggregated across all spatial positions and mode-selected channels by summation, and were separated into positive and negative components prior to aggregation; only positive contributions were retained for visualization. The resulting contribution map was averaged across color channels, median-filtered with a kernel of size 4 to reduce noise, normalized to $[0, 1]$, and scaled by a contrast factor of 7 before being clipped to $[0, 1]$ and used as a multiplicative mask on the mean- and variance-normalized input image for display.

## SUPPLEMENTAL FIGURES

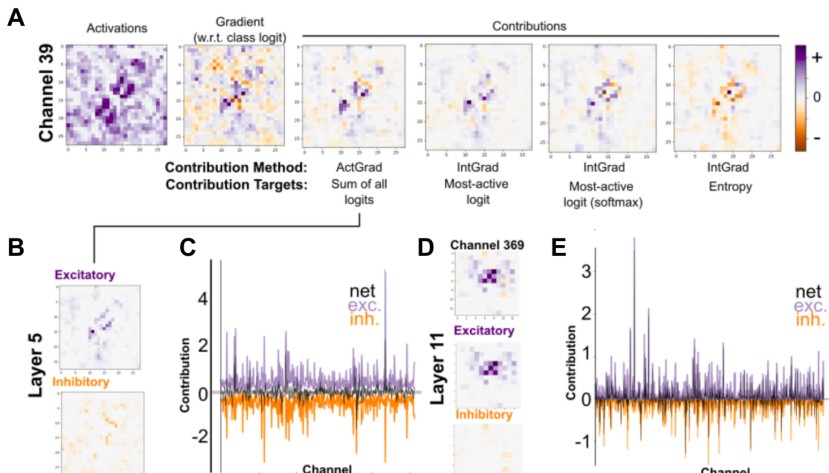

Figure S1: **Comparison of attribution methods and targets reveals consistent channel contribution patterns.** (A) Spatial maps of activations, gradients, and contributions computed using different contribution algorithms (ActGrad, IntGrad) and targets (sum of all logits, most-active logit, most-active logit with softmax, entropy) for Channel 39. All methods show similar spatial sparsity. (B) Excitatory and inhibitory contribution maps for Channel 39 in Layer 5 for a single image demonstrating the separation of excitation and inhibition in an early layer. (C) Contribution values across channels for one image showing the distribution of net excitatory and inhibitory effects for Layer 5. (D) Same as (B) but for Channel 369 in Layer 11, illustrating consistent excitatory/inhibitory patterns across different channels. (E) Same as (C) but for Layer 11, showing how the excitatory/inhibitory decorrelation evolves across network depth.

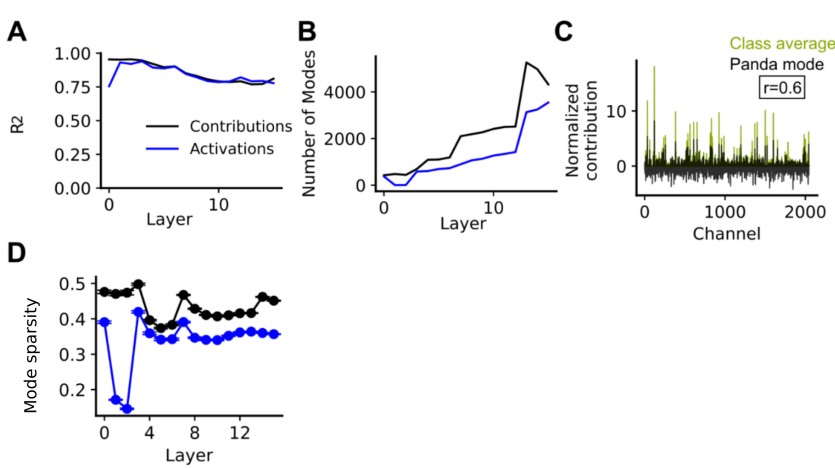

Figure S2: **Sparse autoencoder decomposition performance and sparsity analysis across network layers.** (A) Reconstruction accuracy ($R^2$) of sparse autoencoder decompositions for contributions (black) and activations (blue) across network layers. Both show high reconstruction fidelity, with contributions maintaining slightly higher $R^2$ values in intermediate layers. (B) Number of modes discovered by the sparse autoencoder decomposition as a function of network depth for contributions (black) and activations (blue). The number of modes increases through the network, with contributions yielding more modes than activations in deeper layers. (C) Comparison of the most correlated panda mode with the average contribution pattern during presentation of panda images. The decomposition identified a mode (black) that shows similar contribution patterns to the class-average contributions when panda images are presented (r=0.6). (D) Median Hoyer sparsity index of learned modes across network layers for contributions (black) and activations (blue). Contribution modes maintain higher sparsity than activation modes throughout the network.

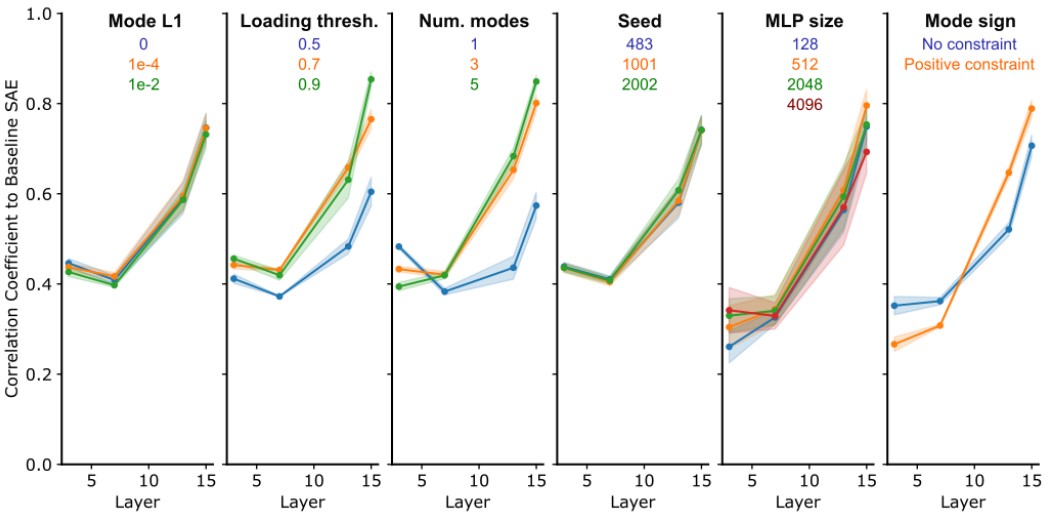

Figure S3: **SAE mode stability across hyperparameters and random seeds.** SAEs were trained on Resnet-50 contributions for example layers 3, 7, 13, and 15, varying hyperparameters L1 sparsity penalty strength, threshold, dictionary size (N * number of channels), random seed, MLP encoder size, and a non-negativity constraint were varied across the indicated values. Each panel shows the correlation coefficient between baseline SAE modes (used in main text) and modes learned with varied hyperparameters as a function of network layer. Modes are aggregated for each class by summing all modes whose correlation with that ImageNet class exceeds 0.2; if no modes exceed this threshold, the single top-correlated mode is used. Shaded regions show mean ± standard error of correlation across 1000 ImageNet classes.

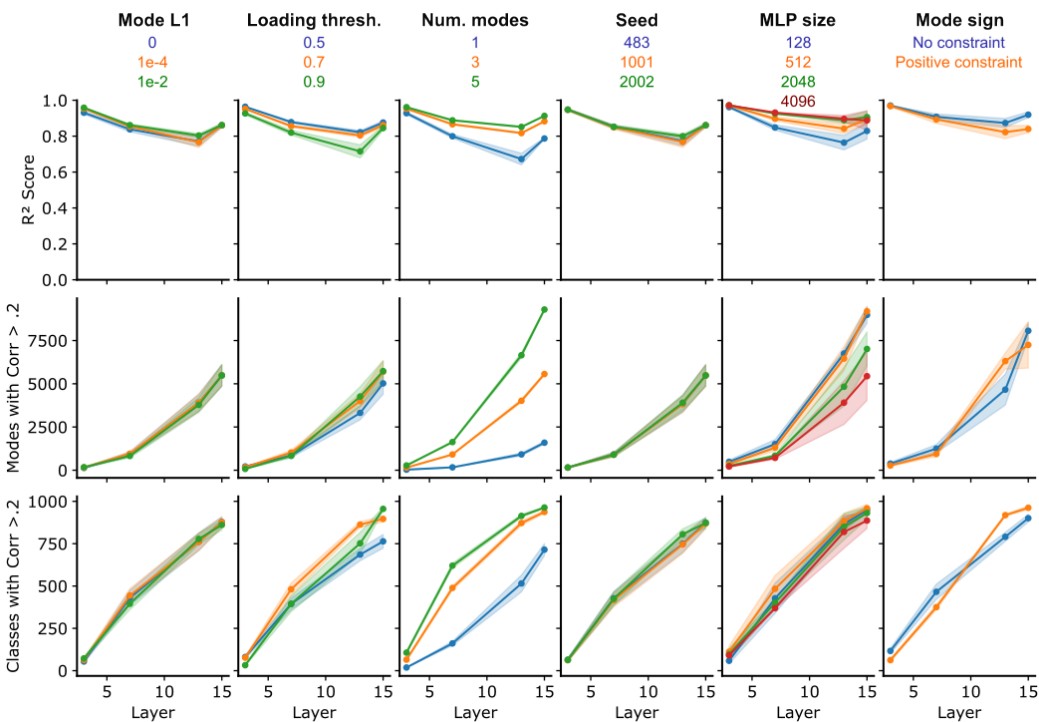

Figure S4: **SAE performance metrics are robust across hyperparameter choices.** Each panel shows a metric (rows) across network layers (x-axis) for different hyperparameter values (colored lines). For each hyperparameter column, values are averaged across all other hyperparameter configurations. Shaded regions indicate mean $\pm$ standard error across configurations. Top row: $R^2$ score, reconstruction quality across all hyperparameters. Middle row: The number of modes having a correlation with classes $> 0.2$ for different hyperparameters. Bottom row: The number of classes having a correlation with modes $> 0.2$ for different hyperparameters.

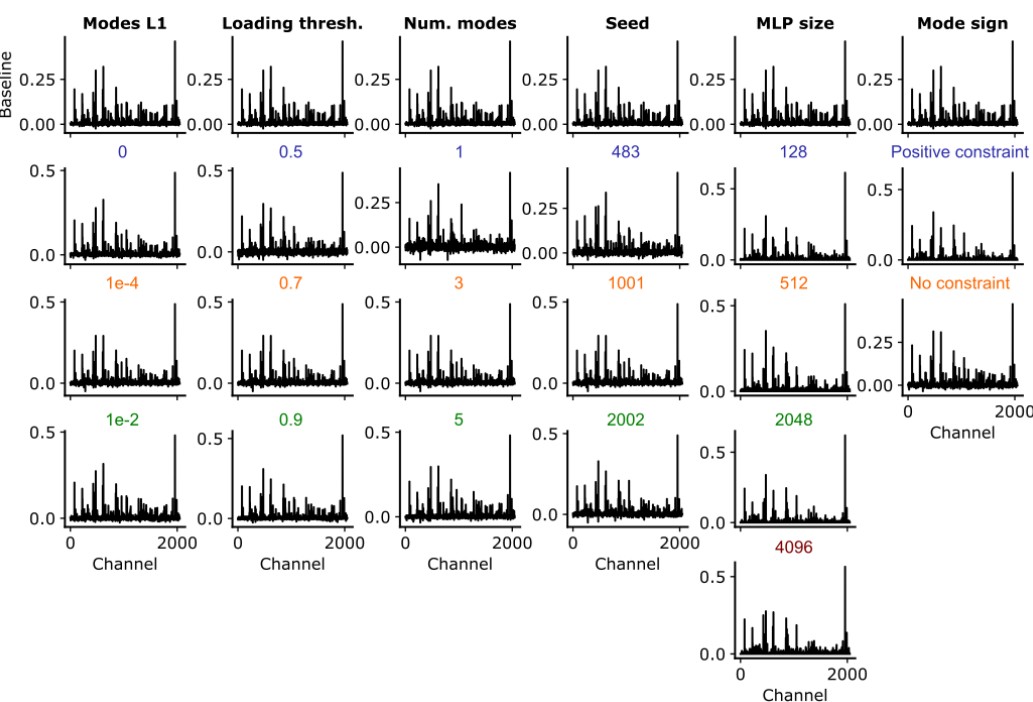

Figure S5: **Class-specific mode representations are consistent across hyperparameter choices.** For the Black Widow class we summed all modes with correlation $> 0.2$ to that class. Shown is the sum for SAEs trained with different hyperparameters.

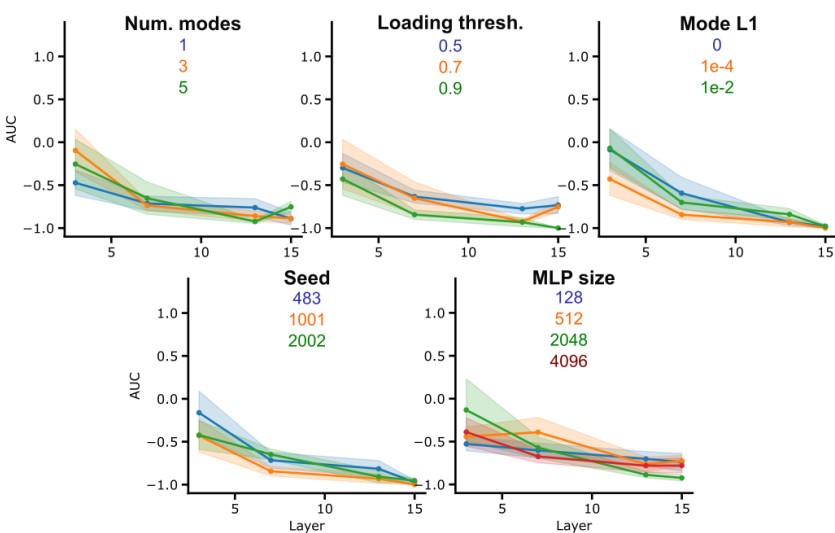

Figure S6: **Hyperparameter robustness of SAE ablation specificity across architectures.** Ablation specificity measured as normalized AUC difference across ResNet-50 blocks (3,7,13,15) for SAEs trained with varying hyperparameters: dictionary size multiplier (N=1,3,5), activation threshold (0.5,0.7,0.9), mode L1 regularization (0, 1e-2, 1e-4), random seed (1001, 2002, 483), and MLP hidden layer size (128-4096). Threshold = 0.9 corresponds to our baseline set of hyperparameters used in the main text. Normalized AUC difference integrates performance ratios (perturbed/original accuracy) across ablation percentages (25%, 50%), with negative values indicating greater specificity (larger target class accuracy drop relative to off-target classes). Results demonstrate robustness across hyperparameter choices, with all configurations showing increased specificity in deeper layers. Error bars indicate SEM across ten trials with randomly sampled class pairs.

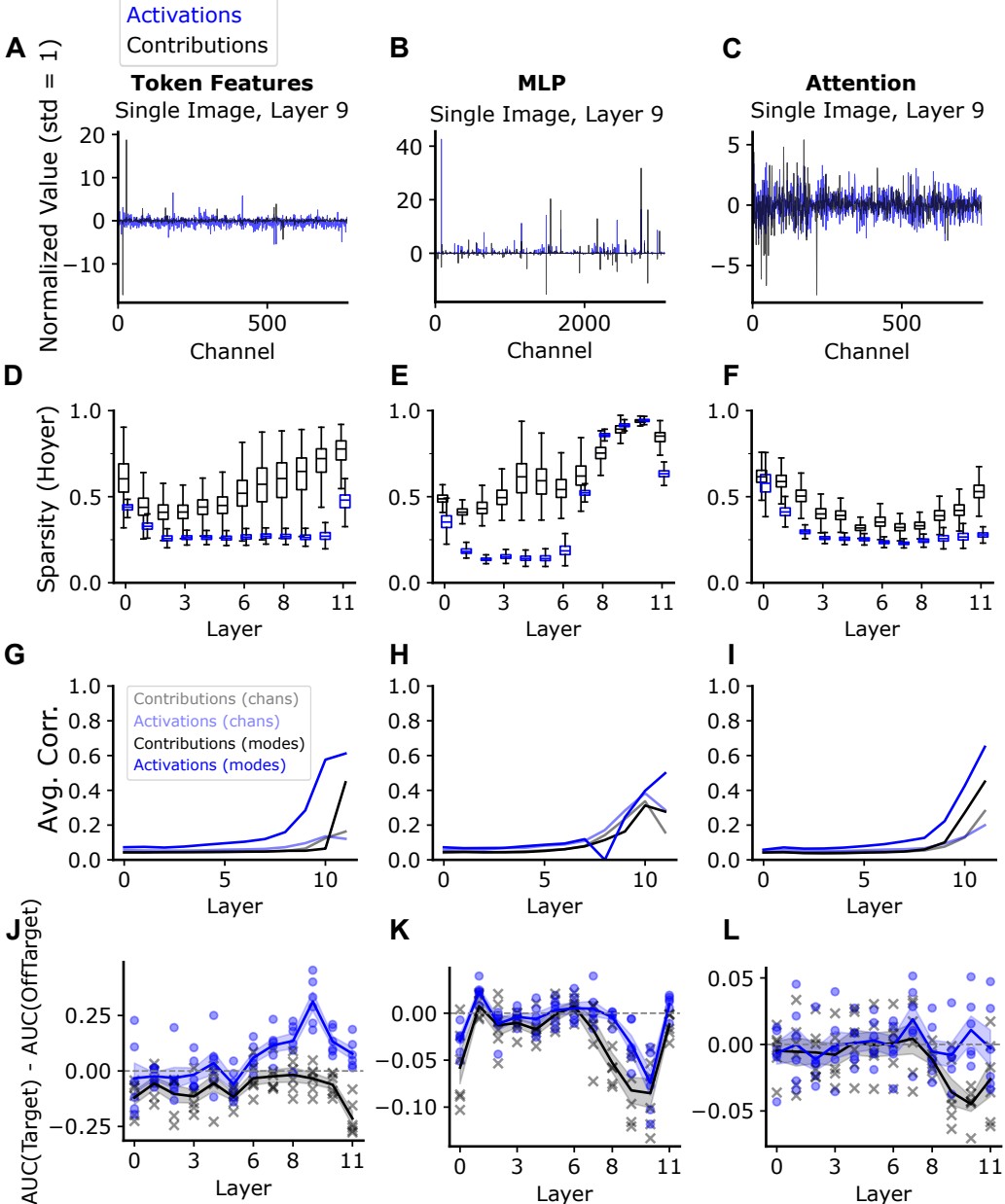

Figure S7: **Application of CODEC on ViT-B.** The columns correspond to the three layer types—token features, MLP, and attention. (A–C) Examples of contributions and activations computed for a single image at layer 9. (D–F) Hoyer sparsity as a function of depth for activations (blue) and contributions (black). Box plots show the distribution of sparsity values over the entire ImageNet validation set, omitting outliers for clarity. (G–I) Analogous to Fig. 5D in the main text, the mean over units (pseudo-channels or modes) of the maximal class-correlation for each unit as a function of depth for four cases: contributions of single pseudo-channels, activations of single pseudo-channels, contribution modes, and activation modes. (J–L) Analogous to Fig. 6E, performance score quantifying perturbation effectiveness as the area between target and off-target curves from Fig. 6D normalized to the off-target area, as a function of depth. Six trials are performed for each layer, shown as scatter points, and the SEM is shown as the shade.

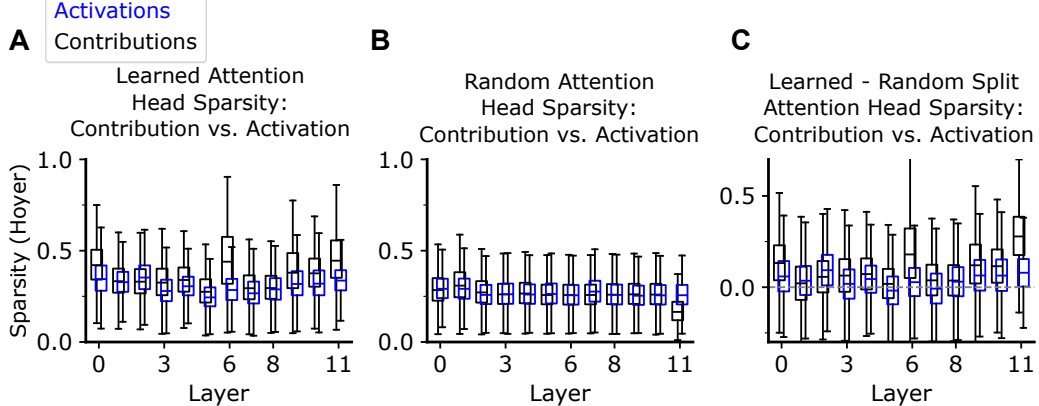

Figure S8: **Attention head sparsity for ViT-B.** (A) Hoyer sparsity as a function of layer after summing contributions (black) or activations (blue) within each attention head. Box plots show the distribution of sparsity values over the entire ImageNet validation set, omitting outliers for clarity. (B) Hoyer sparsity as a function of layer after summing contributions or activations within attention heads formed by a random grouping of hidden dimensions. (C) Distribution of the difference in Hoyer sparsity for the learned and randomly grouped attention heads as a function of layer after summing contributions or activations within each attention head.

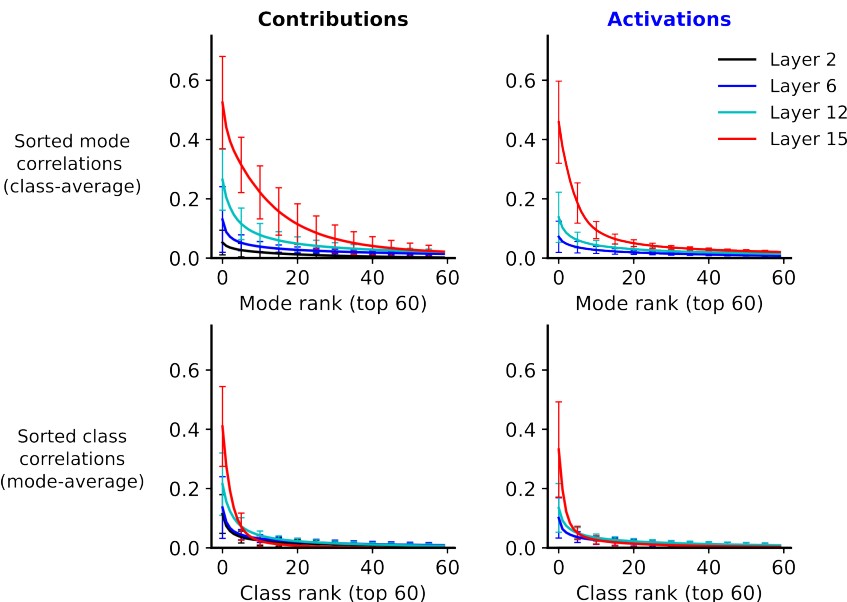

Figure S9: **Contribution modes capture set of dataset classes.** To determine whether our SAE models learned modes that captured the full set of classes in the Imagenet dataset, we computed the full mode-by-class correlation matrix at each layer of the model for both activations and contributions. Sorting this matrix along either the rows or columns gave insights into how many modes individual classes were correlated with (top), and how many classes individual modes were correlated with (bottom). We found that in later layers, classes on average were well covered by learned modes; furthermore, this measure of class-coverage was higher for contribution modes than for activation modes at intermediate layers of the model. Error bars indicate standard deviation.

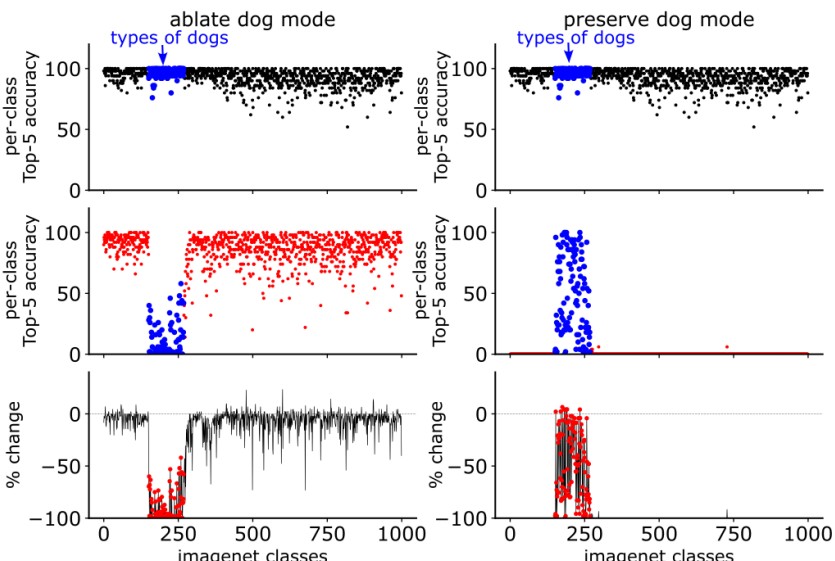

Figure S10: **Manipulation of higher semantic classes using contribution modes.** (A) Ablation experiment: A binary mask was created for all images with dog superclass labels, then correlated with mode loadings to identify dog-correlated modes. All modes with correlation $> 0.2$ were summed together. Removing channels from this aggregated dog mode in intermediate layers (12-15) eliminates the network's ability to classify dog images while leaving other classes largely unaffected. (B) Preservation experiment: Using the same summed dog mode identified through correlation analysis, retaining only these mode channels allows the network to correctly identify dog images while reducing accuracy to near-zero for all other classes.

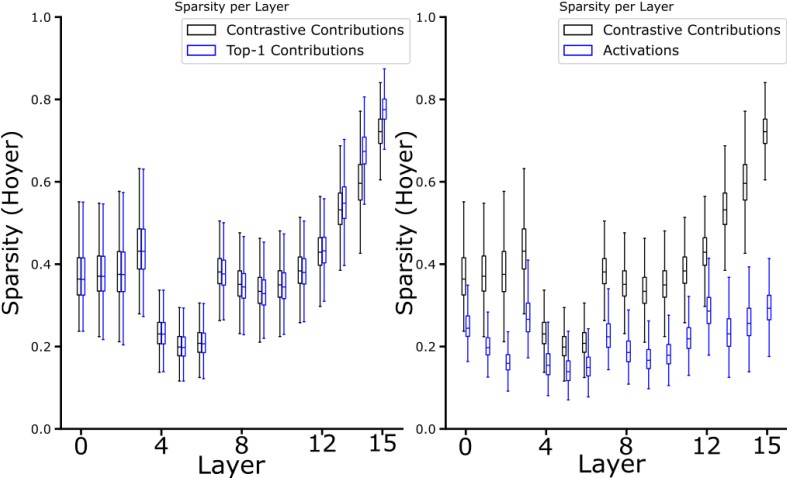

Figure S11: **Layer-wise sparsity comparison of contrastive and top-1 contributions.** Box plots showing the distribution of Hoyer sparsity of contributions across layers 0–15 of ResNet-50 on ImageNet. Black boxes represent contrastive contributions with respect to the difference between top-1 and top-2 predictions, while blue boxes show top-1 contributions with respect to top-1 predictions (left) or activations (right).

