# OpenReview forum: "Causal Interpretation of Neural Network Computations with Contribution Decomposition"
_ICLR.cc/2026/Conference — ICLR 2026 Poster_

### Official Review · Reviewer_Jrdp · 2025-10-23

**Soundness:** 3
**Presentation:** 3
**Contribution:** 2
**Rating:** 6
**Confidence:** 3

**Summary:**

This paper introduces CODEC, an interpretability framework for studying causal contributions of hidden units in artificial neural networks.  In this paper, this is performed by estimating the contribution of hidden units to an output (with ActGrad or Integrated Gradients, for instance), decomposing these contributions into interpretable “modes” with spare autoencoders (SAEs), and then studying the role of different network units (or channels) associated with the learned modes via visualization and/or ablation experiments.  The authors apply this framework to reveal interpretable patterns in CNN image classification models and models of human retinal activity.

**Strengths:**

- To my knowledge, this is the first paper to apply SAEs to contribution matrices instead of model activations.  The authors argue, and methodologically demonstrate, that this can help reveal causal contributions of hidden units that are not revealed through similar analyses on model activations.
- The proposed method is flexible: architecture agnostic, can be adapted to different contribution targets, does not require access to the model’s original training data  (although, see weakness 1). This is a valuable feature of mechanistic interpretability methods.
- Applying CODEC beyond traditional image classification models, the authors reveal interesting insights into casual operations in models of retinal activity.  This type of interpretability for neuroscience models could be quite valuable for NeuroAI researchers seeking to better understand the brain with models that have representationally aligned neural activity.

**Weaknesses:**

- Authors suggest the generality of CODEC to other architectures and tasks but do not provide evidence of this working.  Although logically, it should, it would be beneficial if the authors could substantiate this claim with small experiments on other tasks and/or modalities.
- Human interpretability of modes are not guaranteed and their interpretations are often subjective.  Although in CODEC analyses the authors identify modes that are correlated with class outputs, it is not always the case that SAE modes are interpretable.  The interpretability of the causal contribution of a unit is hinged upon SAE modes being interpretable.
- There is a lack of quantitative evaluation of the coverage (how many classes are represented by distinct contribution modes) or specificity (how uniquely those modes map to a given class).  While the “black widow” and “dog” (appendix Figure 11) examples are illustrative, a broader analysis across all ImageNet categories (or even semantic combination of categories (e.g., “contains animal”),  would clarify whether CODEC consistently identifies compact causal modes, or if such patterns are limited to a subset of classes.

**Questions:**

- How would you position this work alongside other mechanistic interpretability methods based on causal analysis (e.g., Activation Patching [1])?
- The paper highlights the advantages of performing causal analysis on unit contributions instead of activations.  Are there any scenarios where you would expect analysis on the activation space to be more insightful?  When is it a better choice to use CODEC over an alternative, activation-based approach?
- What is the coverage and specificity of CODEC to a broader set of ImageNet classes (see weakness 3)?


[1] Kevin Meng et al. *Locating and Editing Factual Associations in GPT*.

---

> ### Author Response · Authors · 2025-11-21
>
> Thank you for your thoughtful review and for recognizing CODEC’s novelty in applying SAEs to contribution matrices, its methodological flexibility, and its potential value for NeuroAI research.
>
> As stated in the general response, we will provide evidence of CODEC’s applicability by running experiments on ViT-B.
> Comparing contribution to activation analysis, contribution analysis directly answers the question of how groups of hidden units contribute causally to the output as the network operates – contributions measure causality by construction. In contrast, any method analyzing activations must use a secondary step of perturbation and observation to infer causality indirectly, similar to how recording methods in the nervous system alone cannot reveal causal contributions to function. Contributions are a composition of two functions – the activation driven by network input and the gradient to network output. As such, depending on the question, activation analysis or a comparison of activations and gradients, or activations and contributions may be more appropriate. Activation patching breaks down the process of contribution to ask questions about what would happen if the receptive field stage that drove activation were to be exchanged while the projective field stage remained unaltered.
>
> As another example, as addressed in the response to reviewer MSQk, for the question “Do the actions of hidden units combine nonlinearly to drive the output?”, since contributions by construction sum linearly and compose the effects of activations and gradients, contributions alone do not address this question. Similarly, activation analysis alone does not. However, comparing the expectation of linear effects of activations with actual contributions can address this question, which we will include as an analysis.
>
> Overall, contribution analysis addresses mechanistic questions about how hidden units act to produce network function, whereas activation and gradient analyses address lower-level mechanistic questions about how hidden units construct those actions.
> We will include further statistics about the coverage of modes over classes by examining the distribution of sorted correlations between modes and classes. In addition, we have previously spent some time investigating a hierarchy of semantic classes such as animal, etc. We will include some of this data in the next revision.
>
> Regarding human interpretability, you are correct that there are no objective measures of interpretability in the literature, and this is a weakness of the field. As such we include visualizations as examples for subjective evaluation as is done in other work. To aid in this process, we have developed a more complete means of visualizing what information a mode conveys from a specific input to the model output analyzed across different classes. In short, we find that individual modes will highlight discrete concepts in distinct class examples (i.e. hands touching shiny wood, for both images of cello and bow-and-arrow). We will include this updated visualization method in the final version of the paper.

---

### Official Review · Reviewer_MSQk · 2025-10-27

**Soundness:** 3
**Presentation:** 2
**Contribution:** 3
**Rating:** 6
**Confidence:** 4

**Summary:**

This paper introduces CODEC (Contribution Decomposition), a method for interpreting neural networks by analyzing how hidden neurons contribute to network outputs rather than just their activation patterns. The approach uses gradient-based attribution methods (Integrated Gradients, ActGrad) to compute neuron contributions, then applies sparse autoencoders to decompose these contributions into interpretable modes. The authors apply CODEC to ResNet-50 on ImageNet and to models of retinal neural circuits, demonstrating that contribution-based analysis enables better network control and reveals computational structure invisible to activation-based methods.

**Strengths:**

## Strengths

* The paper presents a novel and conceptually appealing perspective by focusing on how hidden neurons contribute to network outputs rather than merely analyzing their activation patterns. This shift is well-motivated from neuroscience principles, where understanding neural function requires examining both receptive fields (input sensitivity) and projective fields (output effects). The theoretical framework is rigorous, with complete input space decompositions properly derived for ActGrad, InputGrad, and Integrated Gradients, providing a mathematically sound foundation for the contribution analysis.

* One particularly compelling aspect of this work is its dual application to both artificial and biological networks. I find the application to retinal neurons especially relevant and valuable, as it represents a novel translation of topics typically associated with mechanistic interpretability of artificial neural networks to models of biological systems. This demonstrates the breadth and potential impact of the CODEC framework beyond standard machine learning interpretability tasks, bridging computational neuroscience and AI interpretability in a meaningful way.

* The paper provides causal validation through ablation and preservation experiments, which convincingly demonstrate that the discovered modes capture functionally relevant channels. These experiments show that targeting specific channels identified by CODEC can selectively eliminate or preserve classification performance for target classes while leaving other classes largely unaffected, providing evidence beyond mere correlation that the method identifies causally important computational pathways.

* The findings themselves are also scientifically interesting. The progressive decorrelation of excitatory and inhibitory effects across network layers is a novel observation that provides insight into how neural networks organize computations hierarchically. Similarly, the observation that contributions become increasingly sparse and high-dimensional through the network offers new perspectives on how feature processing evolves with depth, findings that would not be apparent from activation analysis alone.

**Weaknesses:**

## Weaknesses

* While the paper acknowledges sensitivity to hyperparameters including hidden layer size and regularization, this issue is not systematically studied or addressed. The authors mention that these parameters "may require tuning for different architectures" but provide no guidance on how to perform this tuning, no ablation studies exploring the sensitivity, and no principled approach for selecting appropriate values. This is particularly problematic for the number of modes (k), which appears to be a critical hyperparameter that determines the granularity of the decomposition. Without systematic investigation or clear selection criteria, practitioners would struggle to apply CODEC to new problems.

* The paper lacks any analysis of computational costs, which is a significant omission given that the method involves multiple backward passes for Integrated Gradients integration steps, and additional computation for the sparse autoencoder training and decomposition. Readers cannot assess the practical feasibility of applying CODEC to larger models or compare its computational requirements to alternative interpretability methods. This is especially important given that the authors suggest the method could scale to large language models, yet provide no evidence or analysis of whether this is computationally tractable.

* The choice of contribution targets—specifically the sum of top-k logits and entropy—lacks thorough justification. While these are reasonable choices, the paper does not explain why these particular targets are optimal, whether other targets might be more informative for different analysis goals, or how sensitive the results are to this choice. The single contribution target formulation is motivated primarily by avoiding intractable 3D decompositions, which is a practical consideration but may limit the method's ability to capture class-specific computational pathways.

* Regarding methodological details, the sparse autoencoder training procedure is underspecified in ways that would make reproduction difficult. Critical details such as the optimizer used, learning rate schedules, convergence criteria, weight initialization schemes, and the specific form of sparsity regularization are either missing or relegated to brief mentions. The paper states that results will be made available as supplementary materials, but the main text should provide sufficient detail for readers to understand and evaluate the approach. The lack of these details makes it difficult to assess whether the discovered modes are stable features of the networks or artifacts of particular training procedures.


### Minor Details

* From a presentation standpoint, Figure 8 is particularly difficult to read. The figure attempts to convey complex information about retinal neural network analysis, but most of the labels are too small to be legible, and the overall layout is cramped. I would suggest either significantly enlarging this figure, possibly splitting it across multiple subfigures, or substantially reducing the caption length to allow more space for the visual content.

* Fig 1 is wrongly referenced multiple times: Line 237; Line 262, Line 269

* Missing Fig reference in line 305

**Questions:**

## Questions

* A critical question for practical application of CODEC concerns hyperparameter selection. Can you provide systematic guidance for choosing the number of modes (k), the appropriate sparsity penalties, and other key hyperparameters? Currently, the paper acknowledges that these choices matter but offers no principled approach for setting them. It would be valuable to understand whether there are data-driven methods for selecting k (perhaps based on reconstruction error curves, or information-theoretic criteria), how sensitive the discovered modes are to different sparsity penalty values, and whether certain hyperparameter configurations consistently work well across different network architectures or layers. Without such guidance, practitioners would need to perform extensive manual tuning, which limits the method's accessibility and reproducibility.

* Another important consideration is the stability of the discovered modes. Are the modes stable across different random seeds in the autoencoder training, across different training runs of the original neural network, or across slight architectural variations? This is crucial for understanding whether CODEC identifies fundamental computational structures in the network or whether it is sensitive to arbitrary training details. For instance, if you train ResNet-50 multiple times with different random initializations, do you recover similar modes? If you slightly modify the architecture (for example, changing the number of channels in a layer or using a different residual connection pattern), do the modes remain interpretable and consistent? Understanding this stability would help establish whether modes represent robust computational motifs or are more ephemeral features of particular network instances.

* Finally, to my understanding, the current approach assumes that contributions can be decomposed as linear combinations of modes, which may be a limiting assumption. What about interactions between modes? In practice, neural network computations often involve complex, nonlinear interactions between features, where the effect of one feature depends on the presence or absence of others. The linear decomposition might miss these interaction effects entirely. Have you considered approaches that could capture mode interactions, such as examining joint activation patterns of multiple modes, or using more sophisticated decomposition methods that allow for multiplicative or other nonlinear interactions? Understanding the limitations of the linear assumption and whether important computational structure is being missed would strengthen the interpretation of the results.

---

> ### Author Response · Authors · 2025-11-21
>
> Thank you for your detailed and thoughtful review. We greatly appreciate your recognition of CODEC’s conceptual appeal, rigorous theoretical framework, and the novel application to retinal neural circuits bridging AI interpretability and computational neuroscience.
> We will conduct systematic studies varying the SAE parameters, including sparsity penalty values and hidden layer sizes across multiple layers. We will provide reconstruction error curves and generally provide more information about our SAE training procedure. We will add complete training details including optimizer choice, learning rate schedules, convergence criteria, weight initialization, and exact sparsity regularization formulations to ensure full reproducibility.
>
> We will test mode stability across different autoencoder random initializations. Anecdotally, the channels identified within class-correlated modes are highly consistent - a point we will be sure to include in the supplementary material of the paper.
> With respect to complexity, computing contributions is O(n) in the number of parameters with a constant factor of the number of integration steps to compute the gradient (typically 20, but can be reduced). More efficient methods like ActGrad approximate the complete contribution with only a single forward and backward pass. For training the contribution-SAE, complexity is the same as for activations. In the final version of the paper, we will provide detailed runtime measurements for all components (Integrated Gradients backward passes, SAE training, decomposition) at different scales, and estimate computational feasibility for larger models including preliminary experiments on scaling behavior.
>
> With respect to the stability of the discovered modes across different architectures or network training procedures, we agree these are important, as they are in fact unresolved scientific questions differing in character from issues of robustness to CODEC hyperparameters. These kinds of questions are enabled by approaches like CODEC, and if time permits, it would be interesting to ask whether the modes recovered from activations and contributions on differently initialized CNNs differ in their interpretability and overall alignment.
>
> The question about nonlinear influences raises an important point. Contributions as computed by Integrated Gradients by definition sum linearly, which motivates the linear decomposition of modes. Contributions do not miss nonlinear interactions, but rather they reflect the results of those interactions. For example, if an output C = A * B, the activation A has an integrated gradient of B/2 and the activation B has a gradient of  A / 2, and so the contributions of both A and B are A * B / 2 and sum to the output. Thus the question to ask is not “Are there nonlinear interactions of the modes”, rather it is “Are there nonlinear interactions in the outputs of hidden units reflected in the modes?” This question can be addressed by considering both activations and contributions. One simple method to do this is to compare contribution mode loadings to the expectation of those loadings given the mean gradient (i.e. activation mode * mean gradient), for which we will include statistics.
>
> We will also enlarge Figure 8, fix the Figure 1 references (lines 237, 262, 269) and the missing reference at line 305.

---

> > ### Comment · Reviewer_MSQk · 2025-11-26
> > **Response to Authors' Comments**
> >
> > Thank you for your comprehensive and thoughtful response. I appreciate the substantial commitments you have made to address the concerns raised in my review.
> >
> > I am particularly pleased to see that you will be providing:
> >
> > * Systematic hyperparameter studies including reconstruction error curves and ablations across sparsity penalty values and hidden layer sizes
> > * Complete reproducibility details for the SAE training procedure (optimizer, learning rate schedules, convergence criteria, weight initialization, and exact sparsity regularization formulations)
> > * Detailed computational cost analysis with runtime measurements at different scales and estimates of feasibility for larger models
> > * Mode stability analysis across different autoencoder random initializations, with quantitative evidence supporting your anecdotal observation about consistency
> >
> > Your clarification regarding the nonlinear interactions is helpful, and I look forward to seeing the statistics comparing contribution mode loadings to activation mode × mean gradient expectations. This analysis will be valuable for understanding how well the linear decomposition captures the underlying computational structure.
> >
> > I also appreciate your acknowledgment that questions about mode stability across different network architectures and training procedures represent important unresolved scientific questions. While these go beyond the scope of what should be required for this paper, I would encourage including some discussion of these limitations and their implications for interpreting your results.
> >
> > Based on your commitments, I am willing to raise my score from 6 to 8 pending verification that these additions are included in the revised manuscript. The core contribution of CODEC is valuable and novel, and addressing these methodological and presentation issues will make this a solid contribution to the literature.

---

### Official Review · Reviewer_aiv4 · 2025-10-31

**Soundness:** 3
**Presentation:** 2
**Contribution:** 2
**Rating:** 4
**Confidence:** 4

**Summary:**

The paper introduces a two-stage pipeline that first computes per-channel contribution scores to a scalar target using gradient-based attribution, then trains a sparse autoencoder over these contribution vectors to discover sparse "modes". The authors claim these modes are sparser than activations, align with classes, enable selective output control via targeted channel masking, and support channel-level visualizations. Experiments are on ResNet-50 and a small retinal CNN.

**Strengths:**

* Originality: Clear formulation that swaps activations for contributions before SAE, producing sparse modes that are easy to manipulate
* Quality: Targeted masking yields selective effects for classes inside ResNet-50 and the retina case study shows external face validity
* Clarity: The method is explained step by step, and the visualization procedure is easy to follow
* Significance: The pipeline could be a practical analysis and editing tool for CNNs and may assist mechanistic probing in constrained settings

**Weaknesses:**

* Modern literature already demonstrates SAE-driven feature discovery and steering in ViTs and vision-language models. The submission does not compare against these systems or articulate why contribution-SAE is preferable.
* All main results are in CNNs. There is no validation on ViTs, attention heads, MLP neurons, or token features, which is where much of the community focuses today
* The experiments show interventional control under channel masking in one backbone but do not establish model-invariant causal mechanisms - so the causal language is not entirely warranted
* Emphasis on top-k logits or entropy may conflate classes. Per-logit or contrastive targets would better test class specificity
* No systematic ablation is provided over dictionary width , sparsity penalty, or thresholds. Stability of mode discovery is also unclear
* The "century of neuroscience" narrative should be replaced by a concise, current context with concrete recent citations

**Questions:**

* How does the method extend to ViTs and transformer units? Please report a compact ViT-B experiment at a few layers and compare contribution-SAE against activation-SAE or token features for steering and selectivity.
* Can you evaluate per-logit and contrastive targets and report how mode sparsity, stability, and class selectivity change?
* Please add SAE ablations over dictionary size, L1 strength, thresholding, and signed modeling, with stability metrics for mode-class correlation and intervention effect sizes
* If you select modes across several layers, can you achieve stronger or more selective control than single-layer masking?
* For the retina study, what specific new biological hypothesis or prediction follows from your modes that prior retinal CNN work did not already suggest?

---

> ### Author Response · Authors · 2025-11-21
>
> Thank you for your thorough and constructive review, and for suggestions that will improve the rigor and generalizability of CODEC.
> Towards applying CODEC to another architecture, we have computed contributions from ViT-B at all stages and will run contribution-SAE to provide direct comparisons against activation-SAE to demonstrate relative selectivity and interpretability. From preliminary experiments, contributions are sparser than activations in MLP layers of the transformer and we will add a layer-wise quantification to the final manuscript.
>
> To address the issue of robustness to hyperparameters, we will conduct systematic ablations over dictionary size, L1 penalty strength, thresholding parameters, and the sign of the SAE reporting stability metrics for mode-class correlations and intervention effect sizes. Results will focus on the range across which results are robust.
>
> It is an insightful suggestion to mask channels across multiple layers. We have observed that masking smaller numbers of channels across layers can be a selective and efficient means of ablating a specific concept. We will include examples of this analysis in the next revision.
>
> With respect to computing contributions per-logit or with contrastive targets, we have performed experiments on top-1 logits and the results are consistent. Overall, we will show that the channels identified by contribution-SAE are largely robust to the specific contribution target (multiple top-k, entropy, etc). For example, using the ground truth class-label as the contribution target yields modes that are highly correlated (>0.5) with the modes recovered from the top-k logit approach; we will include these results in the supplemental material. With respect to a contrastive target, note that the entropy measurement is analogous to the top logit contrasted with all other logits. There are various ways to use a contrastive target to answer different questions about what drives class specificity, but we will include results as to predicted vs. runner-up as a common method.
>
> We thank the reviewer for raising an important question about biological predictions. Prior retinal population studies have identified patterns of coactivated retinal neurons that share interneuron presynaptic input  (Prentice et al., 2016, PLoS Comp Biol; Schnitzer & Meister, 2003). In addition, similar groups of cells referred to as retinal ‘modes’ have been proposed to have error-correcting properties (Prentice et al., 2016, PLoS Comp Biol). Because hidden units of retinal CNNs are interpretable in that they have high correlation with retinal interneurons (Maheswaranathan et al, 2023), contribution decomposition automatically generates hypotheses as to which specific interneuron combinations causally construct each mode, and what visual features drive each response. These hypotheses can focus targeted perturbation experiments identifying minimal neuron sets whose joint activity causes specific visual computations. Additionally, our approach could reveal how combinations of neurons with overlapping receptive fields contribute differently to output depending on stimulus context.
>
> With respect to the more general historical neuroscience introduction, we will replace it with recent motivating references of retinal modes and causal perturbations such as electrical stimulation and optogenetics.

---

### Author Response · Authors · 2025-11-21
**All reviewer response**

We thank all reviewers for their thoughtful questions, insights, and excellent suggestions for improvement. We appreciate that all reviewers noted the novelty, practicality, sound motivation, and demonstrated scientific insight. In addition, we appreciate that all reviewers acknowledge the interesting applications of CODEC to both neuroscience and AI. We note that two reviewers (aiv4 and JRDP) point out that we only present results applied to CNNs. CNNs were chosen because they had the closest relationship with the biological system that we analyzed – the retina - and because the method is grounded in principles from visual neuroscience (projective and receptive fields). The method is general, however, and we have already begun applying CODEC to ViT, finding that contributions are also highly sparse compared to activations. We will include analyses of ViT in the final version, and will include results in a revised version before the end of the discussion period. We also note that two reviewers (aiv4 and MSQk) point out the lack of information about robustness with respect to hyperparameters – in the final version of the paper, we will provide a more thorough description of our SAE training procedure and additionally show results from a sweep across hyperparameters (specifically the L1 penalty and the threshold). In brief, the L1 regularization parameter simply influences convergence rate, and the threshold influences sparseness, which can be adjusted by examining reconstruction accuracy. Indeed, the number of recovered modes depends on these hyperparameters; however, the stability of modes that are correlated with semantic classes is anecdotally robust. We will add quantitative justification of this point in the final version of the paper. We respond to all other issues and suggestions below.

---

### Author Response · Authors · 2025-12-03
**Comment Regarding Revised Manuscript**

To briefly orient the AC to our responses and new pieces of supporting material, we wanted to outline what we thought were the most pertinent issues raised by reviewers, and what we have done in order to address them:

Hyperparameter exploration: Nearly all reviewers raised the important point that we did not show how robust or scalable our SAE results were with-respect-to the hyperparameters thereof. We performed significant analyses to answer this question and added a new supplemental section dedicated to this issue. In short, we are able to show that for both ablations and other measures (like sparsity and mode correlations), the original statements from the paper are largely unaffected. Results are insensitive to several hyperparameters, and others only need avoid a bounding region. Overall, contribution modes can be recovered across a wide-range of SAE runs with varied parameters, and these modes converge on the same sets of channels for the same semantic concepts.

VIT: We have successfully implemented CODEC on a standard Vision Transformer architecture—ViT-B. As in CNNs, contributions are more sparse than activations, as they represent the causal effects of activity on network output. Subsequent decomposition reveals contribution modes at various stages of the transformer architecture. We demonstrate that these modes can be leveraged for targeted perturbation experiments comparatively better than those based on activations. We feel that these results are sufficient to show the reviewers that the CODEC method applies to architectures other than convolutional neural networks. Although there are a large set of future interesting analyses to pursue and compare regarding information contained in transformers, our results indicate unique aspects of ViT’s computational strategies including that correlations between activations and target classes show a different progression through the network than correlations between contributions and target classes. We feel that these initial results demonstrate the broad applicability and informative approach of contribution decomposition.

Other additions to the manuscript include a brief runtime summary showing the computational efficiency of CODEC, which we will expand in the final version, a contrastive target, and an improved figure for retinal data. Other items will be included in the final version.

---

### Meta-Review · Area_Chair_5utW · 2026-01-16

**Summary:**

The paper introduces CODEC, a method for interpreting neural networks that uses sparse autoencoders to analyze how hidden neurons contribute to network response.

The paper initially received mixed reviews with two weak accepts and one weak reject. Concerns reported included a lack of validation beyond CNNs, insufficient hyperparameter sensitivity analysis and missing computational cost details. These concerns were addressed in the rebuttal by additional experiments on ViT-B showing generalization to transformers as well as hyperparameter analysis and a quantification of computational cost. This led one reviewer to explicitly state willingness to increase their score to a stronger accept, would likely have caused a second reviewer to maintain their weak accept and the third reviewer to increase their score to a weak accept.

**Reviewer Concerns:**

Concerns reported included a lack of validation beyond CNNs, insufficient hyperparameter sensitivity analysis and missing computational cost details. These concerns were addressed in the rebuttal by additional experiments on ViT-B showing generalization to transformers as well as hyperparameter analysis and a quantification of computational cost.

**Reviewer Scores:**

The rebuttal led one reviewer to explicitly state willingness to increase their score to a stronger accept, would likely have caused a second reviewer to maintain their weak accept and the third reviewer to increase their score to a weak accept.

---

### Decision · Program_Chairs · 2026-01-26

Accept (Poster)